# A friendly introduction to triangular transport

## Abstract

Decision making under uncertainty is a cross-cutting challenge in science and engineering. Most approaches to this challenge employ probabilistic representations of uncertainty. In complicated systems accessible only via data or black-box models, however, these representations are rarely known. We discuss how to characterize and manipulate such representations using *triangular transport maps*, which approximate any complex probability distribution as a transformation of a simple, well-understood distribution. The particular structure of triangular transport guarantees many desirable mathematical and computational properties that translate well into solving practical problems. Triangular maps are actively used for density estimation, (conditional) generative modelling, Bayesian inference, data assimilation, optimal experimental design, and related tasks. While there is ample literature on the development and theory of triangular transport methods, this manuscript provides a detailed introduction for scientists interested in employing measure transport without assuming a formal mathematical background. We build intuition for the key foundations of triangular transport, discuss many aspects of its practical implementation, and outline the frontiers of this field.

## 1 Motivation

**Who is this tutorial for?**   This manuscript is an accessible introduction to *triangular transport*, a powerful and versatile method for generative modelling and Bayesian inference. In particular, triangular transport underpins effective algorithms for data assimilation, solving inverse problems, and performing simulation-based inference, with applications across myriad scientific disciplines.

This tutorial targets researchers with an interest in applied statistical methods but without a formal background in mathematics. Consequently, we will focus more on intuition, general concepts, and implementation, referring the reader to other relevant articles for more formal exposition and theory.

**How does triangular transport work?**   Like other measure transport methods, triangular (measure) transport is a framework to transform one probability distribution into another. This operation is highly useful, as it allows us to characterize a complex target distribution $\pi$ by transforming a simpler, known reference distribution $\eta$. Throughout this manuscript, we use orange to denote variables associated with the (problem-specific) target distribution and green to denote variables associated with the (user-defined) reference distribution, e.g., a standard Gaussian. The idea of coupling two distributions is used in a wide range of applications. In *generative modelling*, for example, we are usually interested in creating samples of a target distribution $\pi$, such as the distribution of $400 \times 400$ pixel images of cats. Measure transport methods approach this challenge by first learning a transport map $\mathbf{S}$ that transforms $\eta$ to $\pi$, and then use $\mathbf{S}$ to convert reference samples $\mathbf{z} \sim \eta$ (here: $400 \times 400$ pixel images of Gaussian white noise) into samples from the target distribution $\mathbf{x} = \mathbf{S}(\mathbf{z}) \sim \pi$ (here: $400 \times 400$ pixel images of cats).

Some of these methods – among them triangular transport – can also characterize *conditionals* $\pi(\boldsymbol{a}|\boldsymbol{b}^*)$ of the *joint* target distribution $\pi(\boldsymbol{a}, \boldsymbol{b})$ of two random variables $\boldsymbol{a}$ and $\boldsymbol{b}$. Here blue denotes the fact that $\boldsymbol{b}^*$ is a deterministic, fixed value. Conditioning operations often arise as stochastic generalizations of evaluating deterministic processes (see Figure 1). As we will describe in Section 2, conditioning is also central to the Bayesian approach to statistical inference, which is an important tool across many scientific disciplines. We

distinguish here between generating from $\pi(\boldsymbol{a}|\boldsymbol{b}^*)$ ("generate an image of a grey cat") and $\pi(\boldsymbol{a}, \boldsymbol{b})$ ("generate a color and a cat of that color") by assuming that $\boldsymbol{b}^*$ is determined outside of our control, either by user or application.

**What makes triangular transport special?** At the heart of triangular transport is their eponymous *triangular* structure. This structure sets them apart from other measure transport methods such as normalizing flows (e.g., Rezende & Mohamed, 2015; Kobyzev et al., 2021), which compose together many simpler but somewhat ad hoc transformations, often interleaved with permutations, and even from conditional normalizing flows (e.g., Van Den Oord et al., 2016), which parameterize normalizing flows in order to represent block triangular (rather than strictly triangular) maps. Triangular structure – discussed in greater detail in subsequent sections – has many important practical properties:

- **Parsimony**: The parameterization of the map function is at the user's discretion (see Section 3.1). This means we can adjust the map's overall complexity, down to the complexity with which it resolves individual variables and variable dependencies. This allows us to implement nonlinear maps that are as complex as necessary, and yet as simple as possible (see Section 4.2).

- **Sparsity**: Triangular maps have a natural ability to exploit *conditional independence*. This improves their computational efficiency, which enables these maps to scale to high-dimensional settings. Further, such structure makes them highly robust to spurious correlations and smaller sample sizes (see Section 2.3.3).

- **Numerical convenience**: Constructing triangular maps boils down to parameterizing simple one-dimensional monotone functions, a task with a rich body of supporting literature. Because of this, these maps are easy to optimize and invert, which we investigate in detail.

- **Explainability**: Triangular maps have a clear correspondence between their constituent elements and the statistical features they represent (see Section 2.3). We can then readily describe different factorizations of the target distribution using the elements of a triangular map, with a particular focus on combinations of various conditional and marginals of the target.

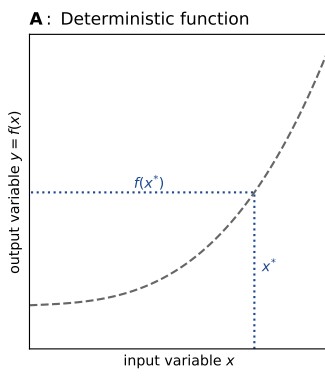
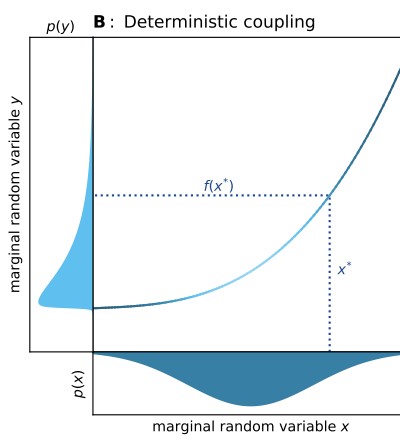
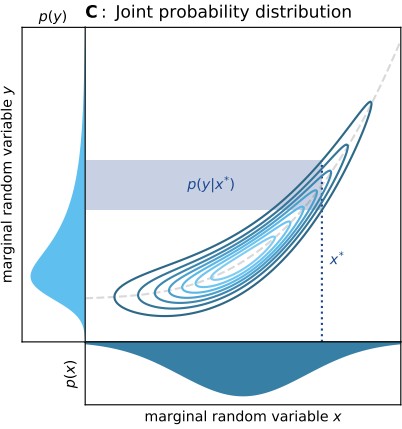

Figure 1: Progression from a fully deterministic to a fully stochastic system. (A) Numerical models are usually represented as deterministic functions. (B) In the presence of input uncertainty, deterministic functions encode a deterministic coupling which yields uncertain output (see Section 2.2). (C) If the function itself is uncertain, this coupling "blurs" into a joint probability distribution. Function evaluation now corresponds to characterizing a conditional distribution. Mind that (A) and (B) can also be parsed as degenerate joint probability distributions.

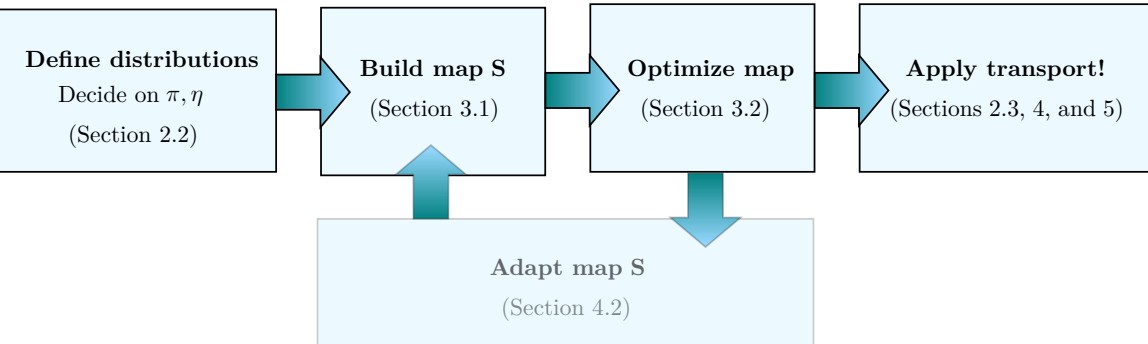

Figure 2: Flowchart of the main steps required to define, build, optimize, and apply triangular maps, with links to the relevant sections.

**In what applications has triangular transport been successful?** Triangular transport has been applied to a wide range of statistical problems in many different disciplines. Examples of such applications include:

- **Bayesian inference**: Triangular transport lends itself exceptionally well to the sampling of conditional distributions. As such, it has found application in both variational (El Moselhy & Marzouk, 2012) and simulation-based inference (Marzouk et al., 2017; Rubio et al., 2023; Baptista et al., 2024a), for large-scale inverse problems (Brennan et al., 2020) and in applications with multiscale structure (Parno et al., 2016).

- **Data assimilation**: Triangular transport provides true nonlinear generalizations of popular filtering (Spantini et al., 2022) and smoothing (Ramgraber et al., 2023a;b) algorithms such as the ensemble Kalman filter and smoother, and their many variants (Grange et al., 2024).

- **Density estimation and generative modelling**: The coupling learned by triangular transport is highly useful for the estimation (Wang & Marzouk, 2022; Martinez-Sanchez et al., 2024; López-Marrero et al., 2024) and sampling (Irons et al., 2022) of non-Gaussian probability distributions, even in high dimensions (Katzfuss & Schäfer, 2024).

- **Optimal experimental design**: Due to the close connections between conditional densities and expected information gain or mutual information, triangular transport maps are useful for estimating common objectives in Bayesian optimal experimental design (Huan et al., 2024; Koval et al., 2024; Li et al., 2024).

Further applications of triangular transport include methods for joint state-parameter inference in state-space models (Spantini et al., 2018; Grashorn et al., 2024; Zhao & Cui, 2024), solving Fokker–Planck equations (Zeng et al., 2023), stochastic programming (Backhoff et al., 2017), and even the discovery of causal models from data (Akbari et al., 2023; Xi et al., 2023).

**How is this tutorial structured?** In the following, we will provide an intuition-focused introduction to the theoretical basics of triangular transport (Section 2), discuss practical aspects related to their implementation in code (Section 3), and conclude with a brief overview of interesting research directions (Section 5). For researchers interested chiefly in practical implementation, a flow chart of the most important steps in the construction and application of triangular transport is provided in Figure 2. First, we define the target $\pi$ and reference $\eta$ (Section 2.3). Then, we structure, parameterize (Section 3.1), and optimize (Section 3.2) the triangular map. Finally, we can deploy the map in the application of our choice, with some practical heuristics listed in Section 4. The tutorial will involve several recurring variables, which are summarized in Table 1.

Table 1: Recurring notation and variables.

| | | | | |
|---|---|---|---|---|
| **bold font** | vector-valued variable or function | | Roman font | scalar-valued variable or function |
| $\mathbf{S}$ | Target-to-reference map | | $S_k$ | $k$-th map component function |
| $\mathbf{R}$ | Reference-to-target map (see Section 3.2.2) | | $\tau$ | twice Archimedes' constant, i.e., 6.283185... |
| $\pi$ | target distribution of interest | | $\eta$ | reference distribution, often standard Gaussian |
| orange | variable associated with $\pi$ | | green | variable associated with $\eta$ |
| $\mathbf{x} \sim \pi$ | target random variable | | $\mathbf{z} \sim \eta$ | reference random variable |
| $\mathbf{X}^i$ | $i$th realization of $\mathbf{x} \sim \pi$ | | $\mathbf{Z}^i$ | $i$the realization of $\mathbf{z} \sim \eta$ |
| $S^\sharp \eta$ | pullback distribution | | $S_\sharp \pi$ | pushforward distribution |
| $K$ | number of target dimensions | | $N$ | ensemble size |
| $p$ | generic probability density function (pdf) | | $\mathbf{a}, \mathbf{b}$ | generic random variables |
| $\mathbf{y}^*$ | conditioning variable | | $\mathbf{x}^*$ | conditioned variable $\mathbf{x}^* \sim p(\mathbf{x}|\mathbf{y}^*)$ |
| $c$ | basis function coefficient | | $r$ | rectifier ($r : \mathbb{R} \to \mathbb{R}^+$) |
| $f$ | monotone function | | $g$ | nonmonotone function |

## 2 Theory

### 2.1 Bayesian inference

To begin, let us briefly revisit some basic concepts of Bayesian inference which will serve to motivate the operations explored in the following sections. In short, Bayesian inference is based on *Bayes' theorem*: given two random variables (RVs) $\mathbf{a}$, $\mathbf{b}$ with joint probability density function (pdf) $p(\mathbf{a}, \mathbf{b})$, we see

$$p\left(\mathbf{a}|\mathbf{b}^*\right) = \frac{p\left(\mathbf{a}\right) p\left(\mathbf{b}^*|\mathbf{a}\right)}{p\left(\mathbf{b}^*\right)}, \tag{1}$$

Equation (1) subsumes three sequential operations (see Figure 3; e.g., Gelman et al., 2013):

1. First, the marginal prior $p\left(\mathbf{a}\right)$ is combined with a conditional observation model (sometimes also called *likelihood model*) $p\left(\mathbf{b}|\mathbf{a}\right)$, yielding a joint probability distribution $p\left(\mathbf{a}, \mathbf{b}\right) = p\left(\mathbf{a}\right) p\left(\mathbf{b}|\mathbf{a}\right)$ over all possible combinations of $\mathbf{a}$ and $\mathbf{b}$. This joint distribution describes how the variable of interest $\mathbf{a}$ and the predicted observations $\mathbf{b}$ relate to each other.

2. Next, this joint distribution is conditioned on a specific observation $\mathbf{b}^*$. In practice, this means evaluating $p\left(\mathbf{a}, \mathbf{b}\right)$ for all possible values $\mathbf{a}$ while keeping $\mathbf{b}$ fixed at the value of $\mathbf{b}^*$. This extracts a slice $p\left(\mathbf{a}, \mathbf{b}^*\right)$ of this joint distribution at $\mathbf{b}^*$ along different values of $\mathbf{a}$.

3. Since the probability densities along this slice do not generally integrate to 1, this slice does not constitute a valid pdf. The final step thus normalizes the probability densities against the slice's probability mass $p\left(\mathbf{b}^*\right) = \int p\left(\mathbf{t}, \mathbf{b}^*\right) d\mathbf{t}$, yielding the posterior pdf $p\left(\mathbf{a}|\mathbf{b}^*\right)$.

In summary, Bayes' theorem first constructs a joint distribution $p\left(\mathbf{a}, \mathbf{b}\right)$ from a prior $p(\mathbf{a})$ and an observation model $p(\mathbf{b}|\mathbf{a})$, then conditions it on a specific observation value $\mathbf{b}^*$ and re-normalizes. In consequence, one could reformulate Equation (1) equivalently as:

$$p\left(\mathbf{a}|\mathbf{b}^*\right) = \frac{p\left(\mathbf{a}, \mathbf{b}^*\right)}{\int p\left(\mathbf{t}, \mathbf{b}^*\right) d\mathbf{t}}, \tag{2}$$

where $p\left(\mathbf{a}, \mathbf{b}^*\right)$ evaluates the joint pdf $p\left(\mathbf{a}, \mathbf{b}\right)$ for all possible $\mathbf{a}$ while keeping $\mathbf{b}$ fixed at $\mathbf{b}^*$, and the denominator acts as a normalizing constant. This equation, or reformulation thereof, lie at the heart of all Bayesian

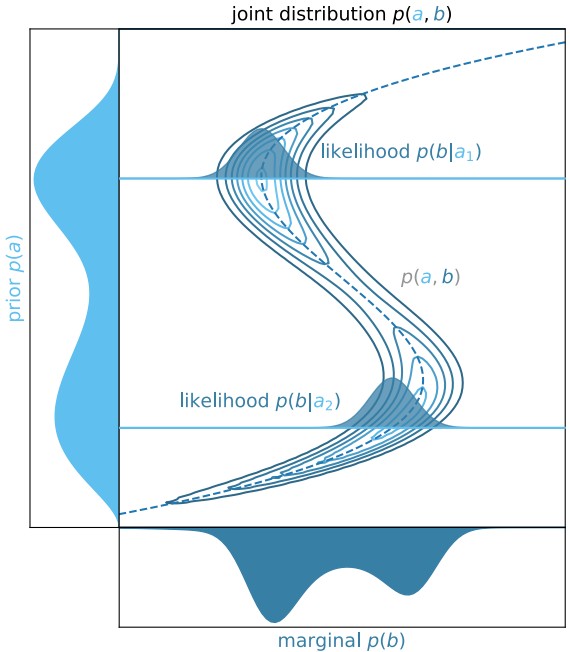
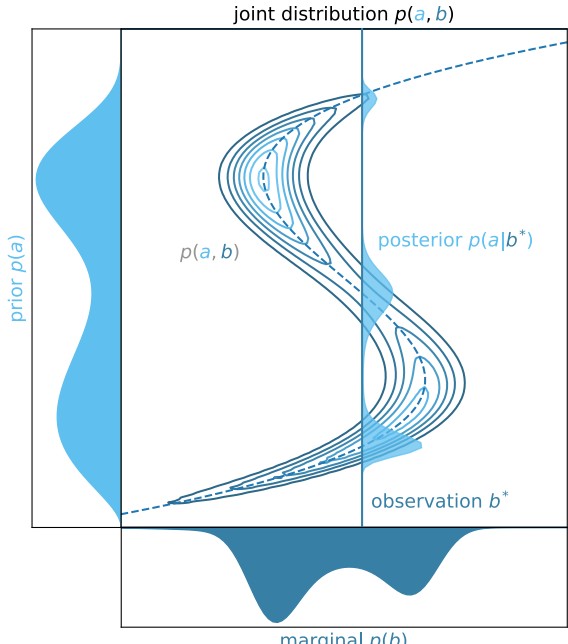

Figure 3: Schematic illustration of Bayes theorem, using a Beta mixture prior $p(a)$ and an observation model $p(b|a) = \mathcal{N}\left(\mu = 2a^3 - a, \sigma = 0.075\right)$. Left: We can create a joint distribution $p(a,b)$ from a prior $p(a)$ and the observation model $p(b|a)$. Right: conditioning this joint distribution on a specific value $b^*$ retrieves a posterior $p(a|b^*)$.

inference algorithms. Unfortunately, it is generally impossible to formulate $p(\mathbf{a},\mathbf{b})$ in closed form, which in turn makes it difficult to evaluate the posterior $p(\mathbf{a}|\mathbf{b}^*)$. To overcome this challenge, different Bayesian inference methods use different strategies. As we shall see in the following, triangular transport solves this challenge by first approximating an almost arbitrary joint pdf $p(\mathbf{a},\mathbf{b})$ by using the concept of measure transport (Section 2.2.1). Crucially, this transformation then allows us to evaluate any of its conditionals $p(\mathbf{a}|\mathbf{b}^*)$ (Section 2.3.2).

The remainder of this tutorial drops this generic notation for two RVs $\mathbf{a}$ and $\mathbf{b}$ jointly distributed as $p(\mathbf{a},\mathbf{b})$ in favour of a single RV $\mathbf{x}$ distributed according to a distribution $\pi$. You can think of $\mathbf{x}$ as an augmented RV $\mathbf{x} = [\mathbf{b},\mathbf{a}]$, and of the joint density as $\pi(\mathbf{x}) = p(\mathbf{a},\mathbf{b})$. Transport methods generally operate on this joint distribution $p(\mathbf{a},\mathbf{b})$; we focus primarily on the ones that will allow us to characterize the desired conditionals $p(\mathbf{a}|\mathbf{b})$.

## 2.2 The change-of-variables formula

The key to understand transport methods is the *change-of-variables formula*. This formula allows us to relate a RV $\mathbf{x}$ associated with a complicated target pdf $\pi$, known only to proportionality or through samples, to a second RV $\mathbf{z}$, associated with a much simpler, user-specified reference pdf $\eta$ through an invertible, differentiable transformation $\mathbf{S}$. In essence, the change-of-variables formula describes what happens to pdfs when they are subjected to specific transformations. For scalar-valued RVs $x$ and $z$, the change-of-variables formula is defined as

$$\pi(x) = S^\sharp \eta(x) = \eta(S(x)) \left| \frac{\partial S(x)}{\partial x} \right|,$$

$$\eta(z) = S_\sharp \pi(z) = \pi(S^{-1}(z)) \left| \frac{\partial S^{-1}(z)}{\partial z} \right|,$$

(3)

where $z = S(x)$ and $x = S^{-1}(z)$, and the *pullback density*[1] $S^\sharp \eta(x)$ obtains $\pi$ by applying the inverse map $S^{-1}$ to the reference distribution $\eta$. The alternate form in the second line of Equation (3) reflects the fact that since $S$ is invertible, each distribution $\pi$ and $\eta$ can be expressed in terms of the other through either the (forward) map $S$ or its inverse $S^{-1}$. Consequently, the *pushforward pdf*[2] $S_\sharp \pi(z)$ likewise approximates $\eta$ by applying the forward map $S$ to the target[3] $\pi$. For multivariate RVs $\mathbf{x}$ and $\mathbf{z}$, the same principle applies. Given a multivariate monotone function $\mathbf{S}$, Equation (3) generalizes to:

$$\pi(\mathbf{x}) = \mathbf{S}^\sharp \eta(\mathbf{x}) = \eta(\mathbf{S}(\mathbf{x})) \left| \det \nabla_\mathbf{x} \mathbf{S}(\mathbf{x}) \right|$$
$$\eta(\mathbf{z}) = \mathbf{S}_\sharp \pi(\mathbf{z}) = \pi(\mathbf{S}^{-1}(\mathbf{z})) \left| \det \nabla_\mathbf{z} \mathbf{S}^{-1}(\mathbf{z}) \right|$$

(4)

The change-of-variables formula in Equation (3) has a surprisingly intuitive interpretation. It states that the probability density $\eta(z)$ at a point of interest $z$ after the transformation equals the original probability density $\pi(x)$ at the pre-transformation point $x = S^{-1}(z)$, adjusted for any deformation the transformation might have induced at this location $|\partial S(x)/\partial x|$. Equation (4) extends this notion to multi-dimensional systems. Intuitively, the absolute value of the determinant of the Jacobian of $\mathbf{S}$, namely $|\det \nabla_\mathbf{x} \mathbf{S}(\mathbf{x})|$, measures the inflation/deflation of an infinitesimal volume centered about $\mathbf{x}$ by the map $\mathbf{S}$. If $|\det \nabla_\mathbf{x} \mathbf{S}(\mathbf{x})| = 1$, the map $\mathbf{S}$ preserves infinitesimal volumes about $\mathbf{x}$. The compensation term, similar to '$u$-substitution' in calculus, is necessary because transformations $\mathbf{S}$ "stretch" or "squeeze" the spaces they are applied to. Accounting for this spatial distortion ensures that the probability mass is preserved and thus the transformed distribution $\eta(\mathbf{z})$ remains a valid pdf.

The change-of-variables formula allows us to describe how a probability distribution $\pi$ changes when subjected to a specific transformation $\mathbf{S}$. An example is provided in Figure 4, which shows how a non-Gaussian target pdf $\pi$ can be related to a Gaussian reference pdf $\eta$ through an invertible transformation; this invertibility is equivalent to monotonicity in the scalar case.

### 2.2.1 Connection to transport methods

The change-of-variables formula has three knobs to tune: the original distribution $\pi$, the map $\mathbf{S}$, and its transformed output $\eta$. When considering the change-of-variables in elementary calculus courses, $\pi$ and $\mathbf{S}$ are often assumed known, and we seek its transformed output $\eta$. Transport methods choose a slightly different approach. We assume $\pi$ and $\eta$ to be known (at least partially), and instead seek the specific map $\mathbf{S}$ which relates the two distributions to each other.

As discussed in Section 1, we generally do not know the target distribution $\pi$ in closed form. Often, the target density $\pi$ is known only partially, either through samples[4] $\mathbf{X} \sim \pi$ or up to proportionality ($\tilde{\pi} = m\pi$, where $m > 0$ is an unknown constant)[5]. On the other hand, the reference distribution $\eta$ is defined as a simple, well-known distribution, often a standard Gaussian pdf $\mathcal{N}(\mathbf{0}, \mathbf{I})$ (Figure 5). The map $\mathbf{S}$ is identified by minimizing an objective function over a specified class of functions, see the discussion in Section 3.2. By finding this map, we learn how to construct the unknown target distribution $\pi$ by transforming a well-defined reference distribution $\eta$.

Among its other uses, learning the map $\mathbf{S}$ allows us to cheaply draw new samples from the target distribution $\pi$. This is achieved by first sampling the reference $\eta$, then applying the inverse map $\mathbf{S}^{-1}$ to the resulting reference samples $\mathbf{z}$. This is especially useful in applications where sampling the target conventionally would involve computationally expensive expensive simulations of, e.g., partial differential equations (*emulation*), or systems in which the sample-generating process is not known exactly (*generative modelling*: e.g., Baptista et al., 2024c).

---

[1] The *pullback* density $S^\sharp \eta(\mathbf{x})$ is the result of applying of the *inverse* map $S^{-1}$ to a RV $z \sim \eta$.

[2] The *pushforward* density $S_\sharp \pi(\mathbf{z})$ is the result of applying the *forward* map $S$ to a RV $x \sim \pi$.

[3] A mnemonic bridge to remember the $\sharp$ notation is that the pull*back* relies on the *inverse* map $\mathbf{S}^{-1}$ to sample, and has the $\sharp$ in the superscript where the $-1$ would be.

[4] Sample approximations are common if $\pi$ is only known from a dataset or if the prior can be sampled but not evaluated.

[5] Unnormalized densities $\tilde{\pi}$ can arise in, e.g., Bayesian statistics, when the prior and likelihood can be evaluated at least point-wise, but it is infeasible to quantify the model evidence (see Equation (1)).

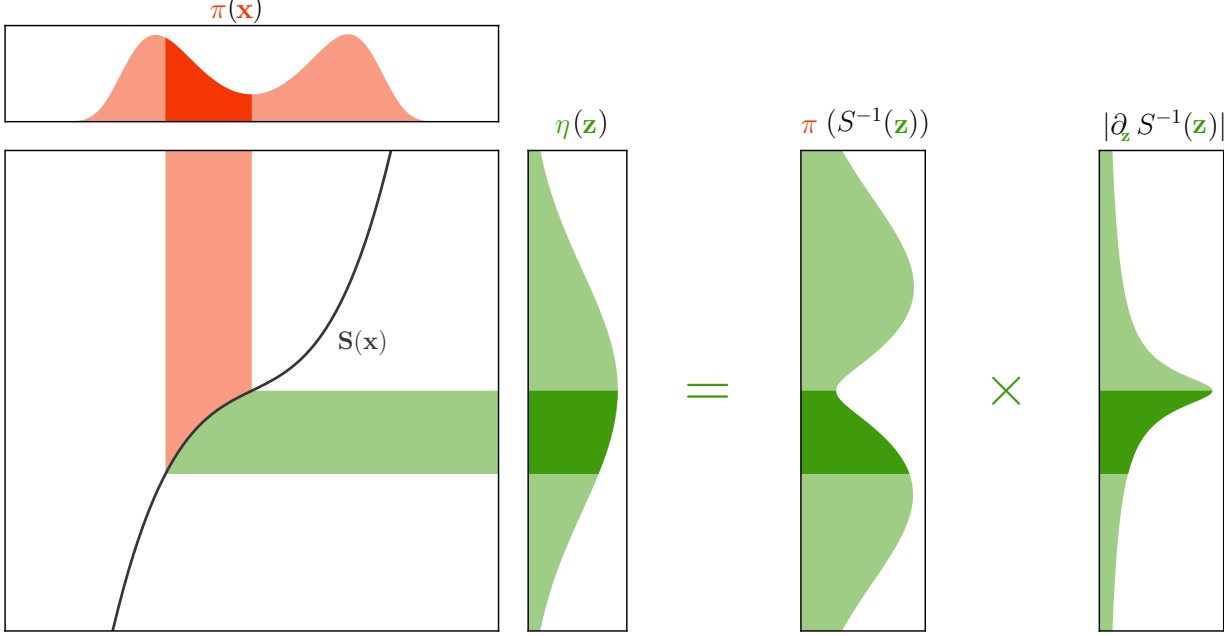

Figure 4: Illustration of the change-of-variables formula. A monotone function $S$ allows us to relate a RV $x$ associated with a pdf $\pi$ to a RV $z$ associated with a pdf $\eta$. We can evaluate $\eta(z)$ by evaluating the target $\pi$ at the pre-image of $z$, that is to say $\pi(S^{-1}(z))$, then multiplying it with the absolute inverse map's gradient $|\partial_z S^{-1}(z)|$.

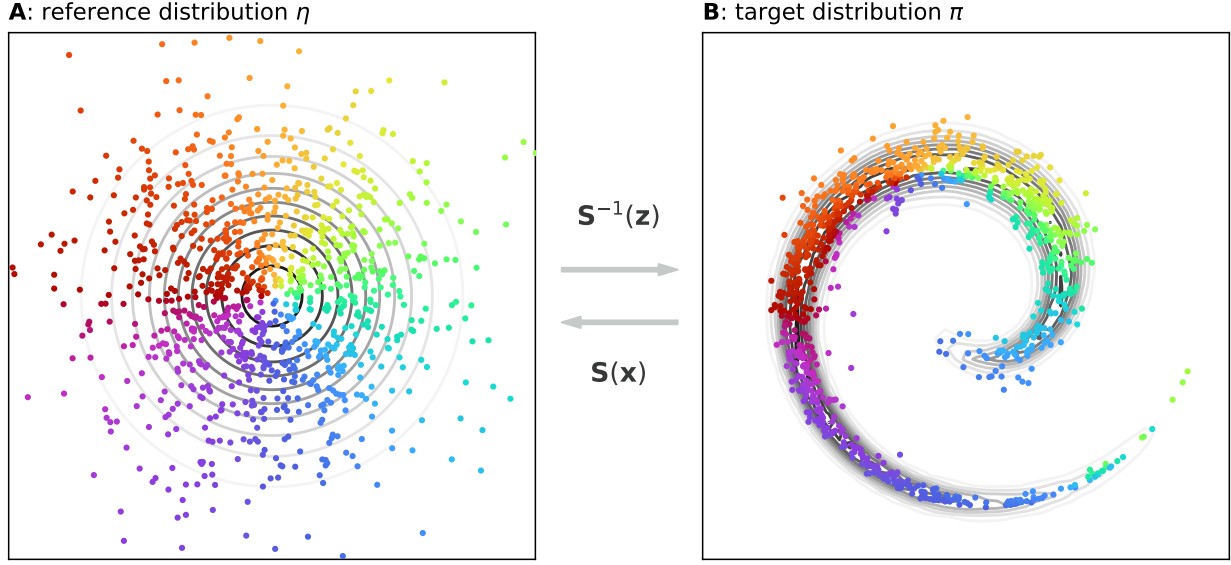

Figure 5: A transport map $\mathbf{S}$ relates a RV $\mathbf{x}$ associated with the target $\pi$ to a RV $\mathbf{z}$ associated with the reference $\eta$. If the map is monotone, it can be applied both ways.

### 2.3   Triangular maps and their uses

Many classes of functions are viable choices for the transport map $\mathbf{S}$. Common examples include normalizing flows and GANs. However, an especially useful class among these are *triangular* transport maps. In addition to sampling the target $\pi$, triangular maps are unique in allowing us to sample *conditionals* of $\pi$, which makes them a flexible and versatile tool for Bayesian inference. Triangular maps are structured as follows:

$$\mathbf{S}(\mathbf{x}) = \begin{bmatrix} S_1(x_1) \\ S_2(x_1, x_2) \\ \vdots \\ S_K(x_1, \ldots, x_K) \end{bmatrix} = \begin{bmatrix} z_1 \\ z_2 \\ \vdots \\ z_K \end{bmatrix} = \mathbf{z}, \tag{5}$$

where the full map $\mathbf{S} : \mathbb{R}^K \to \mathbb{R}^K$ is comprised of map components $S_k : \mathbb{R}^k \to \mathbb{R}$, $k = 1, \ldots, K$, each of which depends only on the first $k$ entries of the target RV $\mathbf{x} = [x_1, \ldots, x_K]^\top$ and we *enforce* that $\partial_{x_k} S_k(x_1, \ldots, x_k) > 0$ for any feasible choice of $x_1, \ldots, x_k$. When all of $S_1, \ldots, S_K$ satisfy this, we call the map $\mathbf{S}$ "monotone". The eponymous *triangular* nature of $\mathbf{S}$ refers to the fact that the map's partial derivatives with regards to $\mathbf{x}$ are lower-triangular; that is to say, the Jacobian matrix $\nabla \mathbf{S}$ has all zeros above its diagonal. This structure—also known as a *Knothe–Rosenblatt rearrangement* (Knothe, 1957; Rosenblatt, 1952) or KR map—has a number of highly desirable properties:

1. Over all functions satisfying this structure, there is one that uniquely couples $\pi$ and $\eta$ under mild conditions (e.g., Marzouk et al., 2017).

2. This triangular structure allows us to evaluate the **determinant of the map's Jacobian** $\det \nabla \mathbf{S}(\mathbf{x})$ efficiently as the product of its diagonal entries (Marzouk et al., 2017), which proves highly useful for the map's optimization (see Section 3.2), not to mention when performing the density estimation itself for the changed variables:

$$\det \nabla \mathbf{S}(\mathbf{x}) = \prod_{k=1}^{K} \frac{\partial S_k(x_1, \ldots, x_k)}{\partial x_k}. \tag{6}$$

3. Triangular maps are **easily invertible**. In particular, we pick a class of functions that are monotone in their last input $x_k$ (with no requirements on the other inputs $x_1, \ldots, x_{k-1}$), i.e. $\partial_{x_k} S_k > 0$ everywhere; this guarantees the monotonicity and thus eases our inversion computation. We will discuss ways to guarantee this property in Section 3.1.1.

4. Perhaps most importantly, triangular maps naturally **factorize the target distribution** into a product of marginal conditional pdfs. We will investigate this property in greater detail in the following sections.

### 2.3.1   Map inversion

In the forward map evaluation (Equation (5)), each of the map's constituent map component functions $S_k$ can be evaluated independently, even in parallel, and then assembled into the full reference vector $\mathbf{z}$. However, the same does not hold for the inverse map:

$$\mathbf{S}^{-1}(\mathbf{z}) = \begin{bmatrix} S_1^{-1}(z_1) \\ S_2^{-1}(z_2; x_1) \\ S_3^{-1}(z_2; x_1, x_2) \\ \vdots \\ S_K^{-1}(z_K; x_1, \ldots, x_{K-1}) \end{bmatrix} = \begin{bmatrix} x_1 \\ x_2 \\ x_3 \\ \vdots \\ x_K \end{bmatrix} = \mathbf{x}. \tag{7}$$

Here, the inverse map's *component functions* $S_k^{-1}$ must be evaluated in sequence and cannot be evaluated independently. This process begins by inverting the first map component $S_1^{-1}(z_1)$, a trivial one-dimensional root finding problem[6], which yields $x_1$. This output $x_1$ serves as auxiliary input for the second map component's inversion $S_2^{-1}(z_2; x_1)$, yielding another one-dimensional root-finding problem, which provides $x_2$. All subsequent map component inversions are similar one-dimensional root-finding problems that likewise depend on the outcomes of previous inversions. This dependence of each inversion $S_k^{-1}$ on each of $x_1, \ldots, x_{k-1}$, the outcomes of previous inversions, effectively factorizes the target distribution as a product of marginal conditionals (Villani, 2007):

$$\pi(\mathbf{x}) = \underbrace{\pi(x_1)}_{S_1^{-1}(z_1)} \underbrace{\pi(x_2|x_1)}_{S_2^{-1}(z_2;x_1)} \underbrace{\pi(x_3|x_1,x_2)}_{S_3^{-1}(z_3;x_1,x_2)} \ldots \underbrace{\pi(x_K|x_1,\ldots,x_{K-1})}_{S_K^{-1}(z_K;x_1,\ldots,x_{K-1})}, \tag{8}$$

where each term corresponds to, and is in turn sampled by, one of the inverse map components indicated in the underbraces[7]. In other words, for a particular sample $i$, each row of Equation (7) can be used to generate a sample $\mathsf{X}_k^i$ from a particular marginal distribution conditioned on $\mathsf{X}_1, \ldots, \mathsf{X}_{k-1}$:

$$\mathsf{X}_k^i = S_k^{-1}(\mathsf{Z}_k^i; \mathsf{X}_1^i, \ldots, \mathsf{X}_{k-1}^i) \sim S_k^\sharp \eta_k = \pi(x_k|x_1, \ldots, x_{k-1}), \tag{9}$$

where $S_k^\sharp \eta_k$ is the pullback of the one-dimensional marginal reference $\eta_k$.

### 2.3.2 Sampling conditionals

As it turns out, the factorization of the target distribution $\pi$ in Equations (8) and (9) also allows us to sample *conditionals* of $\pi$, including the Bayesian posterior $p(\mathbf{a}|\mathbf{b})$ (assuming $p := \pi$ and $[\mathbf{b}, \mathbf{a}] := [\mathbf{x}_{1:k}, \mathbf{x}_{k+1:K}]$). This can be achieved by manipulating the inversion process. First, observe that the factorization in Equation (8) can be aggregated into two blocks:

$$\pi(\mathbf{x}) = \underbrace{\pi(\mathbf{x}_{1:k})}_{\mathbf{S}_{1:k}^{-1}(\mathbf{z}_{1:k})} \underbrace{\pi(\mathbf{x}_{k+1:K}|\mathbf{x}_{1:k})}_{\mathbf{S}_{k+1:K}^{-1}(\mathbf{z}_{k+1:K}; \mathbf{x}_{1:k})}. \tag{10}$$

Similarly, we can aggregate the map component functions into two blocks:

$$\mathbf{S}^{-1}(\mathbf{z}) = \begin{bmatrix} \mathbf{S}_{1:k}^{-1}(\mathbf{z}_{1:k}) \\ \mathbf{S}_{k+1:K}^{-1}(\mathbf{z}_{k+1:K}; \mathbf{x}_{1:k}) \end{bmatrix} = \begin{bmatrix} \mathbf{x}_{1:k} \\ \mathbf{x}_{k+1:K} \end{bmatrix} = \mathbf{x}. \tag{11}$$

Instead of evaluating of Equation (7) from top to bottom ($S_1^{-1}$ to $S_K^{-1}$), if we are interested in sampling conditionals, we may skip the upper map block $\mathbf{S}_{1:k}^{-1}$ and replace its corresponding output $\mathbf{x}_{1:k} = [x_1, \ldots, x_k]$ with arbitrary, user-specified values $\mathbf{x}_{1:k}^* = [x_1^*, \ldots, x_k^*]$. This results in the following truncated inversion for the lower map block $\mathbf{S}_{k+1:K}^{-1}$,

$$\mathbf{S}_{k+1:K}^{-1}(\mathbf{z}_{k+1:K}; \mathbf{x}_{1:k}^*) = \begin{bmatrix} S_{k+1}^{-1}(z_{k+1}; \mathbf{x}_{1:k}^*) \\ S_{k+2}^{-1}(z_{k+2}; \mathbf{x}_{1:k}^*, x_{k+1}^*) \\ \vdots \\ S_K^{-1}(z_K; \mathbf{x}_{1:k}^*, x_{k+1}^*, \ldots, x_{K-1}^*) \end{bmatrix} = \begin{bmatrix} x_{k+1}^* \\ x_{k+2}^* \\ \vdots \\ x_K^* \end{bmatrix} = \mathbf{x}_{k+1:K}^*. \tag{12}$$

Resuming the inversion starting with the map component inverse $S_{k+1}^{-1}$ thus yields samples $\mathbf{x}_{k+1:K}^*$ from the conditional $\pi(\mathbf{x}_{k+1:K}|\mathbf{x}_{1:k}^*)$ instead of the target $\pi(\mathbf{x})$. Equivalent to Equation (8), the corresponding conditional distribution now factorizes as:

---

[6]This root finding problem has a single unique solution within the domain of $S_k^{-1}$, since we require $S_k$ be monotone in $x_k$.

[7]This is only the case if the reference distribution $\eta$ has no dependence between any of its marginals (Spantini et al., 2022), i.e., $\eta(\mathbf{z}) = \eta_1(z_1)\eta_2(z_2)\ldots\eta_K(z_K)$; the standard Gaussian fulfills this property.

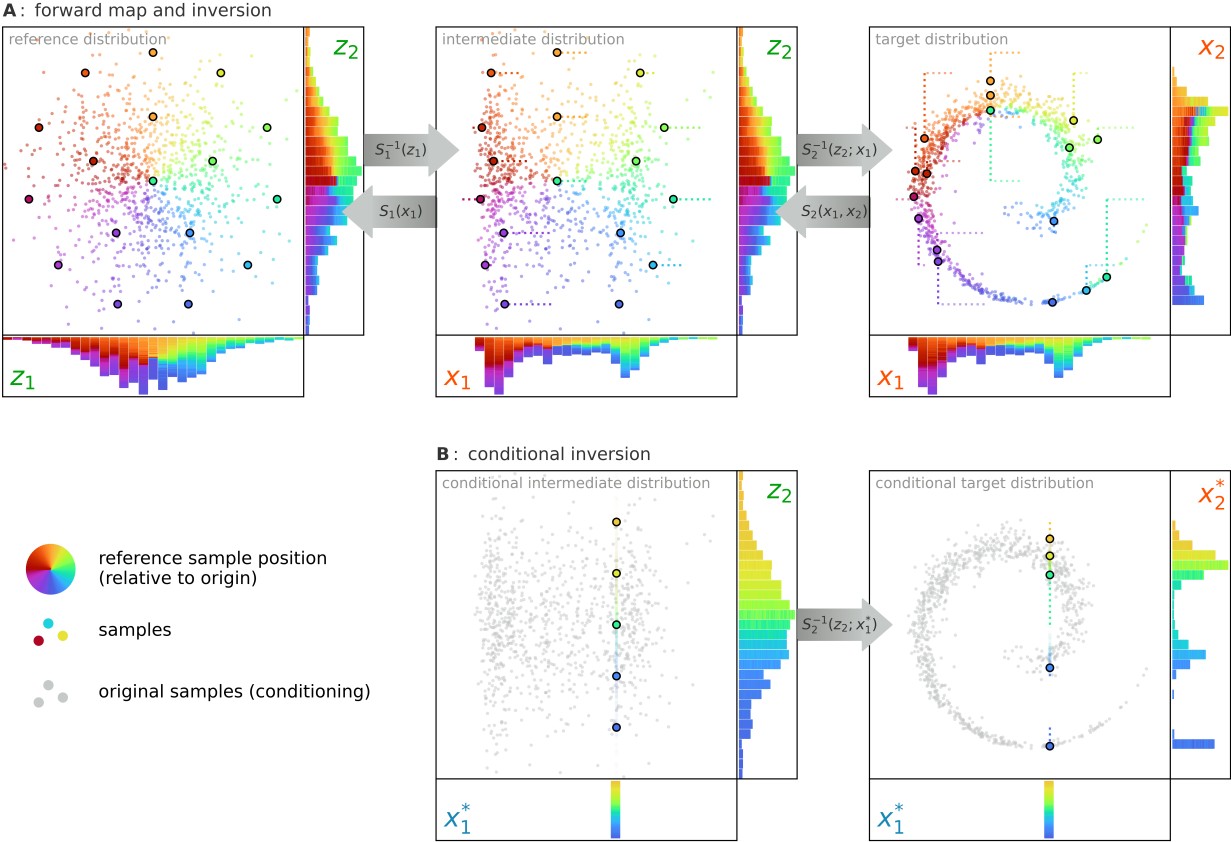

Figure 6: (A) The forward map (top row, from right) and its inverse (top row, from left) operate via implicit intermediate distributions (center), as the map components transform the distributions one marginal at a time. (B) Supplying the inverse map with a manipulated intermediate distribution (bottom row, center) as a starting point will instead sample conditionals of the target distribution.

$$\pi\left(\mathbf{x}_{k+1:K}|\mathbf{x}_{1:k}^*\right) = \underbrace{\pi\left(x_{k+1}|\mathbf{x}_{1:k}^*\right)}_{S_{k+1}^{-1}(z_{k+1};\,\mathbf{x}_{1:k}^*)} \underbrace{\pi\left(x_{k+2}|\mathbf{x}_{1:k}^*,x_{k+1}\right)}_{S_{k+2}^{-1}(z_{k+2};\,\mathbf{x}_{1:k}^*,x_{k+1}^*)} \cdots \underbrace{\pi\left(x_K|\mathbf{x}_{1:k}^*,\mathbf{x}_{k+1:K-1}\right)}_{S_K^{-1}(z_K;\,\mathbf{x}_{1:k}^*,x_{k+1}^*,\ldots,x_{K-1}^*)}, \tag{13}$$

This means that the manipulated triangular map inversion in Equation (7) allows us to sample conditionals of the target distribution $\pi$ for arbitrary $\mathbf{x}_{1:k}^*$, or to estimate the density of this conditional distribution. Recalling Section 2.1, it is plain to see why this operation proves extremely useful in Bayesian inference: If we define our target pdf $\pi$ as the joint distribution $p\left(\mathbf{a}, \mathbf{b}\right)$ (using the notation of Section 2.1) between the RV of interest $\mathbf{a}$ and the observation predictions $\mathbf{b}$, and consider the manipulated samples $\mathbf{x}_{1:k}^*$ to be the observations $\mathbf{b}^*$ of $\mathbf{b}$, then the manipulated inversion in Equation (12) samples the posterior $p\left(\mathbf{a}|\mathbf{b}^*\right)$. An illustration of the forward, inverse, and conditional mapping operations is provided in Figure 6.

### 2.3.3 Conditional independence

A related, very useful property of triangular transport maps is that they naturally allow for the exploitation of *conditional independence*[8] by construction. Recalling the map's generic factorization of the conditionals of

---

[8]By default, many statistical methods assume that all RVs directly influence each other. *Conditional independence* arises whenever two RVs $\boldsymbol{a}$ and $\boldsymbol{c}$ only affect each other indirectly via a third RV $\boldsymbol{b}$. In this case, we say that $\boldsymbol{a}$ is conditionally independent of $\boldsymbol{c}$ (and vice versa) given $\boldsymbol{b}$, represented symbolically as $\boldsymbol{a} \perp\!\!\!\perp \boldsymbol{c} \mid \boldsymbol{b}$. As an example, consider $a$ as the chance of rain, $b$ as the chance of wet ground (which can, but does not have to be caused by rain), and $c$ as the chance of slipping. As rain ($a$) only affects the chance of slipping ($c$) via wet ground ($b$), we have $a \perp\!\!\!\perp c \mid b$.

target $\pi$ in Equation (8), we might ask ourselves what happens in systems in which we can exploit conditional independence. For example, if we have a target distribution $\pi(\mathbf{x}_{1:4})$ and conditional independence properties $x_3 \perp\!\!\!\perp x_1 | x_2$ and $x_4 \perp\!\!\!\perp x_1, x_2 | x_3$ (corresponding to *Markov structure*), the map's factorization could be reduced as follows:

$$
\begin{aligned}
\pi(\mathbf{x}_{1:4}) &= \pi(x_1)\,\pi(x_2|x_1)\,\pi(x_3|\cancel{x_1}, x_2)\,\pi(x_4|\cancel{x_1}, \cancel{x_2}, x_3) \\
&= \pi(x_1)\,\pi(x_2|x_1)\,\pi(x_3|x_2)\,\pi(x_4|x_3).
\end{aligned}
\tag{14}
$$

We refer to this reduction as *sparsification*. Triangular transport maps allows us to leverage these conditional independence properties by simply dropping the corresponding arguments from the map components $S_k$:

$$
\mathbf{S}(\mathbf{x}_{1:4}) = \begin{bmatrix} S_1(x_1) \\ S_2(x_1, x_2) \\ S_3(\cancel{x_1}, x_2, x_3) \\ S_4(\cancel{x_1}, \cancel{x_2}, x_3, x_4) \end{bmatrix} = \begin{bmatrix} S_1(x_1) \\ S_2(x_1, x_2) \\ S_3(x_2, x_3) \\ S_4(x_3, x_4) \end{bmatrix} = \begin{bmatrix} z_1 \\ z_2 \\ z_3 \\ z_4 \end{bmatrix} = \mathbf{z}_{1:4}.
\tag{15}
$$

Equivalently, its inverse map would be:

$$
\mathbf{S}^{-1}(\mathbf{z}_{1:4}) = \begin{bmatrix} S_1^{-1}(z_1) \\ S_2^{-1}(z_2; x_1) \\ S_3^{-1}(z_3; \cancel{x_1}, x_2) \\ S_4^{-1}(z_4; \cancel{x_1}, \cancel{x_2}, x_3) \end{bmatrix} = \begin{bmatrix} S_1^{-1}(z_1) \\ S_2^{-1}(z_2; x_1) \\ S_3^{-1}(z_3; x_2) \\ S_4^{-1}(z_4; x_3) \end{bmatrix} = \begin{bmatrix} x_1 \\ x_2 \\ x_3 \\ x_4 \end{bmatrix} = \mathbf{x}_{1:4}.
\tag{16}
$$

Making use of conditional independence properties in this way is useful for two reasons:

1. **Robustness**: Any conditional independence we can enforce by construction is statistical information the map does not have to pry from the samples or the model, improving the overall fidelity and robustness of the approximation to $\pi$ in settings with finite ensemble size.

2. **Efficiency**: The removal of superfluous dependencies reduces the number of input arguments to many of the map components $S_k$, decreasing the evaluation and inversion complexity (e.g., see Section 3.1.1 for evaluation complexity). This property is often called sparsification as it turns the Jacobian $\nabla \mathbf{S}$ into a sparse matrix.

This second property is the key to applying transport methods in high-dimensional systems. As each map component function $S_k$ generically depends on all previous arguments $\mathbf{x}_{1:k-1}$, the computational demand explodes with the dimension of the target $\pi$ when optimizing or evaluating the map. With sufficient conditional independence, however, sparse maps can overcome this dramatic increase in complexity. For instance, the Markov structure in Equations (15) and (16) results in a sparse map $\mathbf{S}$ with numerical complexity scaling linearly in the target dimension $K$.

A comprehensive account of the link between conditional independence and the sparsity of triangular maps is given in Spantini et al. (2018), through two main lines of results. First, given a sparse undirected probabilistic graphical model that encodes Markov properties of the target distribution $\pi$, it is shown how to predict the sparsity pattern of the triangular map $\mathbf{S}$. This process relies on an ordered graph elimination algorithm, and can thus be performed before learning the map itself. But the resulting sparsity pattern depends on the chosen ordering of the random variables, which underscores the fact that triangular maps are intrinsically *anisotropic* objects: a good ordering is necessary in order to maximize sparsity. We comment further on methods for finding such orderings in Section 5. Second, Spantini et al. (2018) show that a property somehow dual to sparsity is *decomposability*: given the Markov structure of some distribution $\pi$ on $\mathbb{R}^K$, the inverse of the map $\mathbf{S}$ defined by $\mathbf{S}_\sharp \pi = \eta$ can be represented as the composition of finitely many low-dimensional triangular transport maps, where low dimensionality here means that each component map is a function only of a small number of inputs. The exact structure of this decomposition follows from, again, an ordered decomposition of the original graph. These results allow very high-dimensional problems to be broken into many smaller and more manageable parts, given some conditional independence.

### 2.3.4 Block-triangular maps

Recalling the block structure in Section 2.3.2, we have so far assumed that the map component "blocks" in Equation (11) are internally triangular; i.e., $\mathbf{S}_{1:k}$ has output $[S_1(x_1), \ldots, S_k(\mathbf{x}_{1:k})]$ and similar for $\mathbf{S}_{k+1:K}$. We may instead use *any* invertible functions $\mathbf{S}_{1:k} : \mathbb{R}^k \to \mathbb{R}^k$ and $\mathbf{S}_{k+1:K} : \mathbb{R}^K \to \mathbb{R}^{K-k}$, for example certain neural networks (e.g., Baptista et al., 2024c). This still permits sampling conditionals and can be numerically advantageous in high-dimensional systems, but can compromise the ability to exploit conditional independence within the map blocks. In the following, we will assume that the transport map is *fully triangular*, even where we adopt the block-triangular structure for ease of notation.

## 3 Implementation

In the preceding section, we have discussed the theoretical foundations of triangular transport, but stopped short of defining the precise method to evaluate $S_k$, or the optimization problem which identifies the specific map $\mathbf{S}$ from $\pi$ to $\eta$. In this section, we will fill these gaps, and provide guidance on the practical implementation of triangular transport methods in code. First, though, a slight interlude about the "work" a map performs to transform $\pi$ and $\eta$ into one another. Without a loss of generality, suppose $\pi$ and $\eta$ have the same mean and covariance. In the case that both are Gaussian, this implies that they are the same distribution, and the transport map $\mathbf{S}$ is trivially the identity. Once $\pi$ becomes non-Gaussian, the transport map must be strictly nonlinear, for any linear map merely transforms these first two moments! The nonlinearity of a map is therefore directly correlated with how "different" our reference and target distributions are, measured in one way or another (e.g., the Kullback–Leibler divergence, discussed below). This theoretical idea underpins many practical considerations; for example, we will assume that $\pi$ and $\eta$ approximated have the same tails (i.e., $\pi(x) \approx C\eta(x)$ for some $C \in \mathbb{R}$ when $x$ is sufficiently large), implying the map *must* become close to linear sufficiently far from the origin.

### 3.1 Structuring the map components $S_k$

A core component of any triangular transport map $\mathbf{S}$ is the definition of the constituent map components $S_k$. In this subsection, we will introduce and discuss important features in the definition of these map components $S_k$.

#### 3.1.1 Monotonicity

Let us recall from Section 2.2 that each map component function must be monotone in its last argument $x_k$, that is to say, $\partial_{x_k} S_k > 0$ on the *entire* domain (i.e., for any choice $\mathbf{x}_{1:k-1}$). Together with the triangular structure and linearity in the tails, this property guarantees that the resulting map $\mathbf{S}$ is bijective.

This may not be immediately intuitive, so an illustration of this effect is provided in Figure 7, which shows an example in $\mathbb{R}^2$. Here, invertibility ensures that every tuple $(\mathsf{X}_1, \mathsf{X}_2)$ maps to a unique tuple $(\mathsf{Z}_1, \mathsf{Z}_2)$. Graphically, this means that if we overlay Figure 7C and Figure 7D on top of each other, any pair of contour lines between the subplots must not intersect more than once. This is achieved due to triangularity and monotonicity:

- The **triangular structure** ensures that the contours of every map component $\mathbf{S}_{1:k-1}(\mathbf{x}_{1:k-1})$ independent of $x_k$ and are thus constant along $x_k$. In consequence, the contours of $S_1$ in Figure 7C are constant along $x_2$, which aligns them parallel to $x_2$ (see Figure 7C).

- The **monotonicity** of $S_1(x_1)$ relates each $\mathsf{X}_1$ to a unique $\mathsf{Z}_1$. We then obtain a unique tuple $(\mathsf{Z}_1, \mathsf{Z}_2)$ when we ensure that the second set of contours from $S_2(\mathsf{X}_1, x_2)$ increases monotonously along $x_2$ (see Figure 7D). This is achieved through the **monotonicity requirement** $\partial_{x_k} S_k > 0$.

The same principle extends to higher dimensions: the triangular structure means that the $(k-1)$-dimensional contours of $\mathbf{S}_{1:k-1}(\mathbf{x}_{1:k-1})$ are aligned along $x_k$. Monotonicity of $S_k$ along $x_k$ then ensures that for $(\mathsf{X}_1, \ldots, \mathsf{X}_{k-1})$ and any $x_k$, we obtain a unique tuple $(\mathsf{Z}_1, \ldots, \mathsf{Z}_k)$. There are several strategies that can

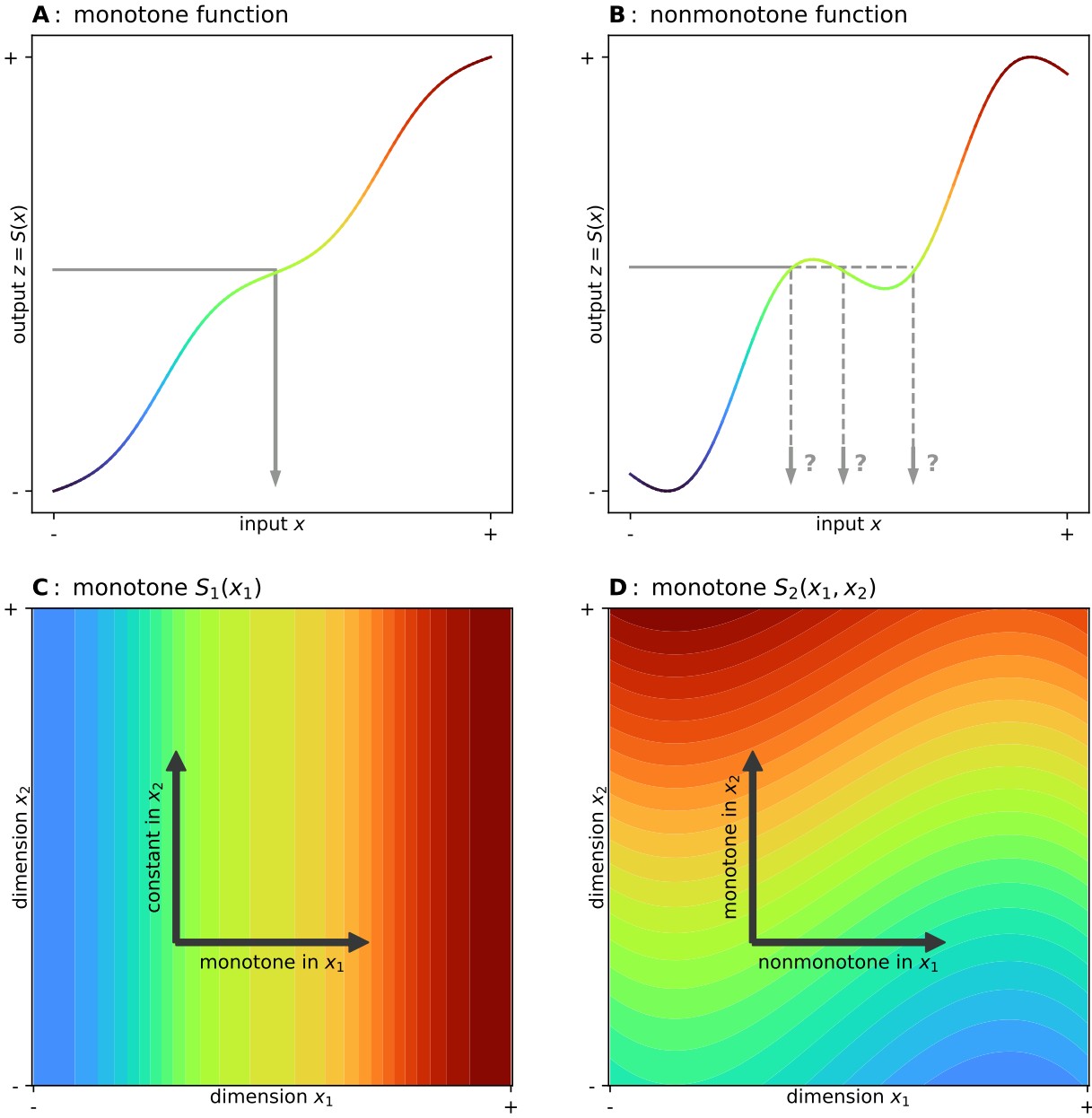

Figure 7: (A) Monotonicity guarantees that a function remains invertible. (B) Non-monotone map component functions have non-unique inverses. How do multivariate maps **S** remain monotone? (C) The map component function $S_1(x_1) = z_1$ is monotone in $x_1$ but constant in $x_2$, as it does not depend on this input. (D) The map component function $S_2(x_1, x_2) = z_2$ is monotone in $x_2$ but can be nonmonotone in $x_1$. Color indicates the magnitude of the map component's output in all subplots.

ensure each map component satisfies these monotonicity requirements. We will explore these strategies in the following.

**Monotonicity through integration**   A general, powerful, but also computationally demanding method to enforce monotonicity makes use of a *rectifier* and *integration* (e.g., Baptista et al., 2023). In this formulation,

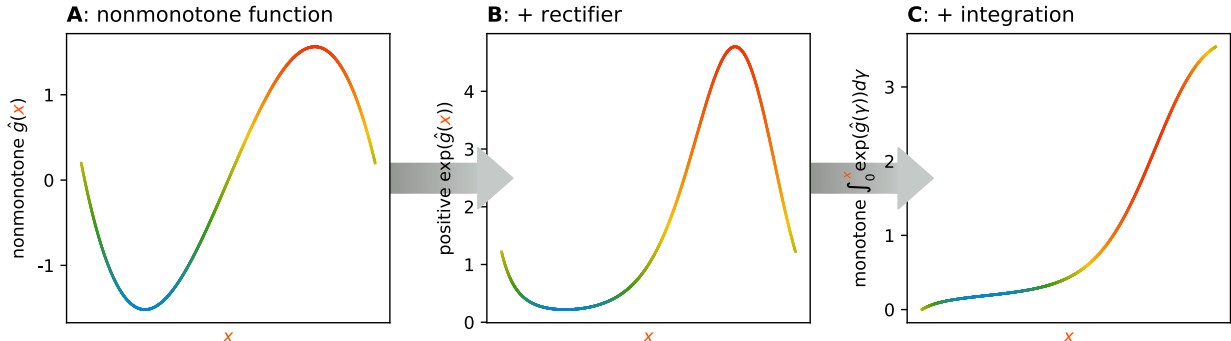

Figure 8: Integrated rectifiers create monotonicity. (A) We start with an arbitrary smooth, non-monotone function $\hat{g}(x)$. (B) Applying a monotone rectifier, for example $\exp(\hat{g}(x))$, yields strictly positive function output. (C) Integrating a strictly positive function yields a monotone function $f$. Color indicates the magnitude of the nonmonotone function in (A).

our starting point is a smooth, differentiable, but nonmonotone function $S_k^{\text{non}} : \mathbb{R}^k \to \mathbb{R}$. An example of such a nonmonotone function is:

$$S_3^{\text{non}}(x_1, x_2, x_3) = \underbrace{c_0 + c_1 x_1 + c_2 x_2 + c_3 x_1 x_2 + c_4 {x_1}^2}_{\text{nonmonotone part } g(x_1, x_2)} + \underbrace{c_5 x_3 + c_6 x_1 {x_3}^2 + c_7 x_2 x_3}_{\text{pre-monotone part } \hat{g}(x_1, x_2, x_3)}, \quad (17)$$

where the coefficients $\{c_i\}$ parameterize $S_k^{\text{non}}$. In Equation (17),we have conceptually separated the terms of $S_3^{\text{non}}$ into a nonmonotone part $g$, which does not depend on $x_3$, and a "pre-monotone" part $\hat{g}$, which does. In general, $\hat{g}$ will not be monotone in $x_3$, and consequently neither will be $S_3^{\text{non}}$. However, we can monotonize it by first applying a rectifier $r : \mathbb{R} \to \mathbb{R}^+$ to $\hat{g}$, then integrating the rectified output over $x_k$:

$$S_k(x_1, \ldots, x_k) = \underbrace{g(x_1, \ldots, x_{k-1})}_{\text{nonmonotone part}} + \underbrace{f(x_1, \ldots, x_k)}_{\text{monotone part}},$$

$$\text{where} \quad \underbrace{f(x_1, \ldots, x_k)}_{\text{monotone function}} = \int_0^{x_k} \overbrace{r(\underbrace{\hat{g}(x_1, \ldots, x_{k-1}, t)}_{\text{nonmonotone function}})}^{\text{positive function}} \, dt. \quad (18)$$

The two steps combined in Equation (18) are as follows:

1. **Rectification**: First, $\hat{g}$ is sent through the rectifier $r$, a function which maps its output to strictly positive values. Examples of useful rectifiers include exponential functions, the softplus function (i.e., $\log(1 + \exp(x))$) or Exponential Linear Units (Clevert et al., 2016).

2. **Integration**: Numerically or analytically integrating the output of the resulting strictly positive function over $x_k$ yields a function $f$ monotone in $x_k$, which turn also guarantees the monotonicity of $S_k$ in $x_k$.

This process is visualized in Figure 8, and has a number of important advantages. First off, the use of an integrated rectifier places very few limitations on the structure of $S_k^{\text{non}}$. Furthermore, it also permits the introduction of *cross-terms* such as $x_1 x_k$ between the last argument $x_k$ and previous dimensions $x_{1:k-1}$, which are required for many of the more complex mapping operations (see Section 3.1.2). The drawback of this approach is that in practice evaluating Equation (18) often demands one-dimensional numerical integration via, e.g., quadrature, increasing the computational demand of the map's evaluation, optimization, and inversion.

An important practical consideration when using the integrated map formulation is to ensure that the pre-monotone term $\hat{g}$ reverts to a *constant* in the tails of $x_k$. This can be ensured by defining $\hat{g}$ as a combination of a constant term and Hermite functions (see Section 3.1.3), which revert to zero in the tails. To understand why this is important, consider the process illustrated in Figure 8. Through rectification and integration, increasingly negative values of $\hat{g}$ become flat stretches in $S_k$. If one or both tails of the monotone term $f$ become near-flat, it is possible for the root finding during the map's inversion (see Section 2.3.1) to fail for outlying values. However, if $\hat{g}$ instead reverts to a positive constant in the tails, the resulting integrated monotone term $f$ will extrapolate linearly and thus generally remain more robust to outliers during the inversion. This is an artifact from the discussion above where, while $\pi$ and $\eta$ may "look" very different for the bulk of the pdf, we expect the tails of both of them to behave similarly (Baptista et al., 2023).

**Monotonicity through variable separation**    A more computationally efficient way to ensure monotonicity is to formulate a map component function $S_k$ which is both *separable in $x_k$* and *linear in the coefficients*, though not generally linear in the inputs $\mathbf{x}$. An example for such a map component function is provided below:

$$S_3\left(x_1, x_2, x_3\right) = \underbrace{c_0 + c_1 x_1 + c_2 x_2 + c_3 x_1 x_2 + c_4 {x_1}^2}_{\text{nonmonotone part } g(x_1, x_2)} + \underbrace{c_5 x_3 + c_6 \operatorname{erf}(x_3)}_{\text{monotone part } f(x_3)}, \tag{19}$$

where $c_i$ once more are the map component's coefficients, which are optimized to find the map from $\pi$ to $\eta$, and $c_5, c_6 > 0$. The key aspects in Equation (19) are a clear separation into a nonmonotone term $g$, which may depend on all arguments except the last $\mathbf{x}_{1:k-1}$, and a monotone term $f$, which may *only* depend on the last argument $x_k$. To ensure that $S_k$ remains monotone in $x_k$, all terms in $f$ must likewise be monotone basis functions (e.g., $x_k$ and $\operatorname{erf}(x_k)$).

The advantages of this separable formulation are two-fold: First, Equation (19) has no need for numerical integration, which reduces computational demand substantially. Second, as we shall see in Section 3.2, this separable formulation allows for extremely efficient map optimization. The price for this efficiency is that variable separation does not allow for cross-terms between the last argument and previous arguments (e.g., $x_1 x_k$), which limits the map's expressivity. We will explore the consequences of this limitation in the next section.

### 3.1.2   Parameterization

Closely related to the concern of monotonicity is how each map component $S_k$ relates $x_k$, its last argument, to $x_1, \ldots, x_{k-1}$. This decision leads to three different kinds of map parameterizations, which permit maps of different complexity. In ascending order of complexity, these parameterizations are:

**Marginal maps**    If both the nonmonotone part $g$ and the monotone part $f$ depend exclusively on input $x_k$, such map component functions $S_k$ result in a *marginal* (or *diagonal*) map. As the name implies, such maps only transform the marginals of $\mathbf{x}$ without capturing any dependencies between its dimensions. An example of a marginal map would be:

$$S_3\left(x_3\right) = \underbrace{c_0}_{\text{nonmonotone part } g(-)} + \underbrace{c_1 x_3 + c_2 {x_3}^3}_{\text{monotone part } f(x_3)}. \tag{20}$$

Recalling the implications of removing dependencies on earlier arguments $\mathbf{x}_{1:k-1}$ from the map component functions (see Section 2.3.3), marginal maps are of no direct use for the conditioning operations in Section 2.3.2. This is because marginal maps implicitly assume all entries of $\mathbf{x}$ are marginally independent, which in turn implies that $\pi(\mathbf{x}_{k+1:K}|\mathbf{x}_{1:k}) = \pi(\mathbf{x}_{k+1:K})$, that is to say, nothing can be learned from $\mathbf{x}_{1:k}$ about $\mathbf{x}_{k+1:K}$. As marginal Gaussianization schemes, however, they can find limited use in Gaussian anamorphosis (Schöniger et al., 2012; Zhou et al., 2011) or as preconditioning tools in certain graph detection methods (Liu et al., 2009). Marginal maps are only included here for completeness' sake, and we use $g(-)$ to denote that the nonmonotone part is constant in $\mathbf{x}$, connecting better to the more complex parameterizations.

**Separable maps** Closely related to the monotonicity scheme of the same name in the previous section, *separable* maps separate the map component $S_k$ into the sum of a function $g : \mathbb{R}^{k-1} \to \mathbb{R}$ that takes in $\mathbf{x}_{1:k-1}$ and a univariate monotone function $f : \mathbb{R} \to \mathbb{R}$ that just takes in $x_k$

$$S_2\left(x_1, x_2, x_3\right) = \underbrace{c_0 + c_1 x_1 + c_2 {x_1}^2 + c_3 x_1 x_2}_{\text{nonmonotone part } g(x_1, x_2)} + \underbrace{c_4 x_3 + c_5 {x_3}^3}_{\text{monotone part } f(x_3)} . \tag{21}$$

Separable maps are one level of complexity above marginal maps. Such maps can consider nonlinear dependencies between the different dimensions, but do not permit cross-terms with the last argument $x_k$, which in turn limits the complexity of the target distributions $\pi$ they can recover[9]. This limitation is subtle, and will be discussed shortly. An important advantage of separable maps is that, when linear in the coefficients, their optimization can be partially solved in closed-form, which can improve computational efficiency substantially. More detail on this is provided in Section A.

**Cross-term maps** The most versatile map parameterization are *cross-term* maps, which permit the greatest expressiveness at the cost of increased computational demand. As mentioned previously, cross-terms are basis functions which depend both on the last argument $x_k$ and preceding arguments $\mathbf{x}_{1:k-1}$. The presence of these terms requires that such maps must use integrated rectifiers (Section 3.1.1) to ensure the monotonicity of $S_k$:

$$S_2\left(x_1, x_2\right) = \underbrace{c_0 + c_1 x_1 + c_2 {x_1}^2}_{\text{nonmonotone part } g(x_1)} + \underbrace{\int_0^{x_2} \exp\left( \overbrace{c_3 {x_1}^2 t}^{\text{cross-term}} + c_4 t + \overbrace{c_5 x_1 t^2}^{\text{cross-term}} \right) dt}_{\text{monotone part } f(x_1, x_2)} . \tag{22}$$

The presence of cross-terms permits increased control over the local details in the transformations from $\pi$ to $\eta$. In principle, it is also possible to make Equation (22) separable in $x_k$ by removing the dependencies of $f$ on $\mathbf{x}_{1:k-1}$. Since the resulting $S_k$ would not be linear in the coefficients $c$, however, we cannot make use of a more efficient optimization scheme (Section A). In the following subsection, we will develop some intuition about the strengths and limitations of different map parameterizations.

**Choosing a parameterization** To provide more insight into the effects and limitations of each parameterization choice, we have illustrated the pullback $\mathbf{S}^\sharp \eta$ of a transport map approximated using these parameterizations for each of three different target pdfs $\pi$ in Figure 9. In every case, we begin by sampling the target distributions, proceed to optimize the different maps, then apply the inverse map (Section 2.3.1) to transform reference samples $\mathbf{z} \sim \eta$ into samples from the pullback $\mathbf{x} \sim \mathbf{S}^\sharp \eta$. We may make the following observations:

- **Marginal** maps only reproduce the marginal densities of all three target distributions. As their structure implies independence between the marginals (Section 2.3.3), however, the pullback approximation does not approximate the true target pdfs well. This effect can be subtle: in the corner-multimodal distribution, it is only noticeable through the emergence of a fourth phantom mode.

- **Separable** maps provide much better approximations to the target pdfs, but have some interesting caveats: while they successfully recover the wavy target distribution, they struggle with the distribution once it is rotated by 90°, and they yield a slight – if less pronounced – fourth phantom mode for the corner-multimodal target. We will discuss the reason for this below.

- **Cross-term** maps prove most versatile, recovering all three target distributions well at the cost of a larger parameter space.

---

[9]Cross-terms and coefficient linearity are not mutually exclusive. However, one must carefully ensure monotonicity in the final argument for any choice of other inputs $\mathbf{x}_{1:k-1}$

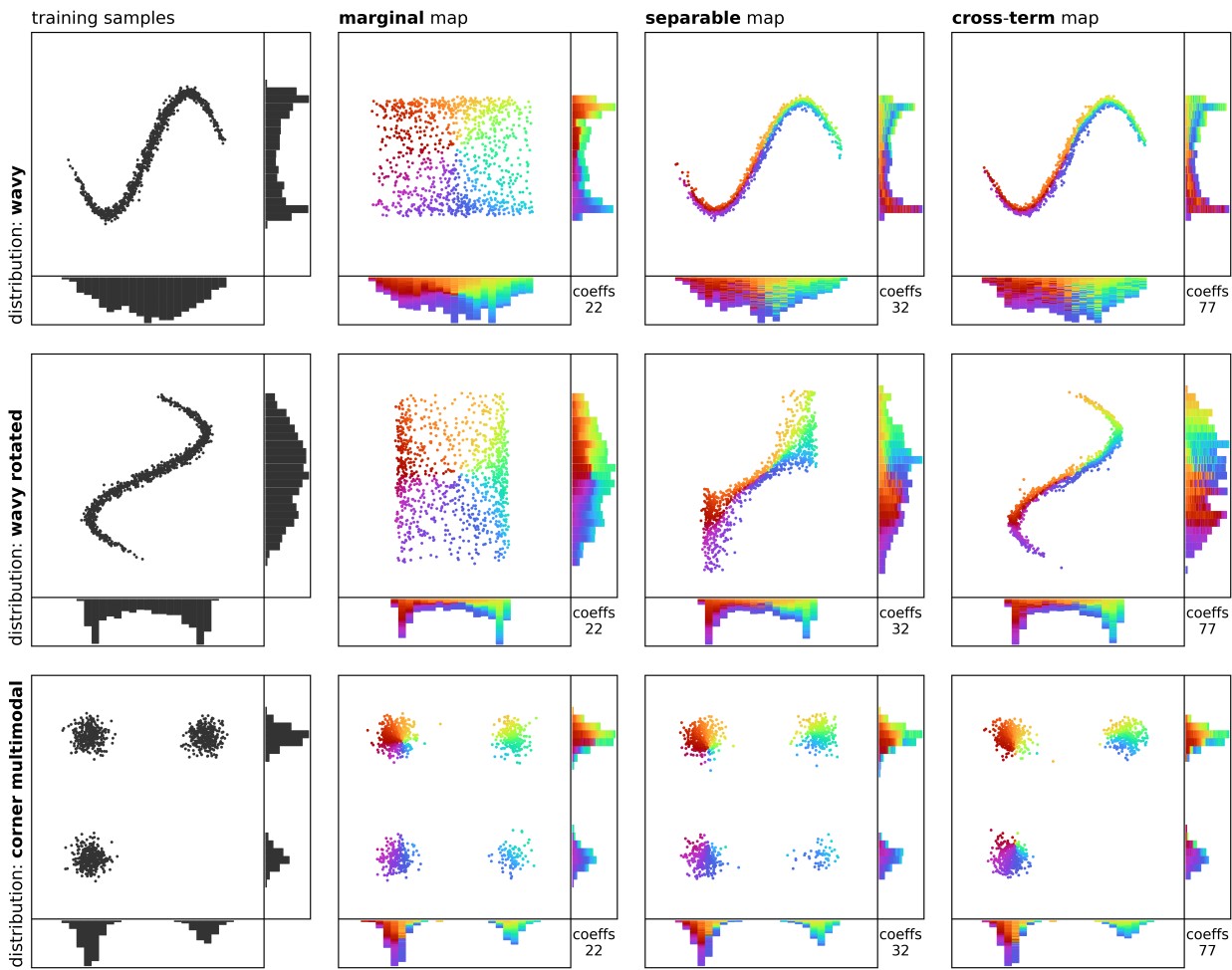

Figure 9: Pullback approximations $\mathbf{S}^\sharp \eta$ to different target distributions $\pi$ using nonlinear map components $S_k$ for marginal maps, separable maps, and cross-term maps. Some distributions require cross-term maps, for others simpler parameterizations may suffice. The variable ordering is $[x_1, x_2]$, with $x_1$ plotted on the horizontal axis and $x_2$ on the vertical axis. Color represents the position of the reference samples $\mathbf{z}$ relative to the mean of $\eta$. The number of coefficients (i.e., parameters) for each map is plotted in the bottom-right corner.

In practice, cross-terms provide greater control over local features. To understand why, let us take a closer look at a single map component inversion $S_k^{-1}$, which forms the basis of the map inversion (see Section 2.3.1) that ultimately yields the pushforward samples in Figure 9. Consider a generic map component function comprised of a nonmonotone part $g$ and a monotone part $f$:

$$z_k = S_k(\mathbf{x}_{1:k-1}, x_k) = g(\mathbf{x}_{1:k-1}) + f(\mathbf{x}_{1:k-1}, x_k). \tag{23}$$

As discussed in Section 2.3.1, the inversion of $S_k$ relies on one-dimensional root finding. Reformulating this expression yields the root finding objective for $S_k^{-1}$, conditioned on the outcomes $\mathbf{x}_{1:k-1}$ of previous map component inversions:

$$z_k - g(\mathbf{x}_{1:k-1}) = f(\mathbf{x}_{1:k-1}, x_k). \tag{24}$$

If we now apply the simplifications of the three map parameterizations above, we obtain three different objectives for the root finding problem:

$$\begin{aligned} z_k - g(-) &= f(x_k) && \text{(marginal maps)} \\ z_k - g(\mathbf{x}_{1:k-1}) &= f(x_k) && \text{(separable maps)} \\ z_k - g(\mathbf{x}_{1:k-1}) &= f(\mathbf{x}_{1:k-1}, x_k) && \text{(cross-term maps)} \end{aligned} \tag{25}$$

These equations provide some insight into the differences between the pullback densities $\mathbf{S}^\sharp \eta$ we have observed in Figure 9. Each term above takes on a different role during the conditional inversion: the reference samples $z_k$ are an independent input, unaffected by the map parameterization choice, and define a *random initial offset* for the root finding. The nonmonotone term $g$ acts as an additional *dynamic offset* for the inversion based on previous values, and the monotone term $f$ defines the *shape of the inverse function* for the one-dimensional root finding over the unknown $x_k$. From this perspective, the differences between the map parameterizations emerge from how each handles the dependence on previous values $\mathbf{x}_{1:k-1}$:

1. **Marginal** maps (Figure 10, top row) permit only a constant nonmonotone term $g(-)$, which results in no dynamic offset. Their monotone term $f(x_k)$ likewise only depends on $x_k$. In consequence, marginal maps extract the same $x_k$ from a specific $z_k$ for any given value of $\mathbf{x}_{1:k-1}$.

2. **Separable** maps (Figure 10, center row) also feature monotone term $f(x_k)$ that depends only on $x_k$, which keeps the inverse function's shape constant for all values of $\mathbf{x}_{1:k-1}$. However, their off-diagonal term $g(\mathbf{x}_{1:k-1})$ can induce varying offsets for different $\mathbf{x}_{1:k-1}$, which introduces a simple dependence on previous values.

3. **Cross-term** maps (Figure 10, bottom row) can vary both the dynamic offset (nonmonotone term $g(\mathbf{x}_{1:k-1})$) and the inverse function's shape (monotone term $f(\mathbf{x}_{1:k-1}, x_k)$) with different $\mathbf{x}_{1:k-1}$, and thus provide the most expressive maps, but often at higher computational cost.

These insights reveal why separable maps succeeded to recover the wavy distribution, but failed to capture its features once it is rotated (Figure 9). For the original wavy distribution, its (vertical) conditionals at different horizontal positions are mostly of the same shape, just shifted vertically, which makes them a perfect fit for recovery via separable maps. Once the distribution is rotated, however, the true vertical conditionals become more challenging to recover. Note that if we were to change the order of the target vector to $[x_2, x_1]$ instead of $[x_1, x_2]$, separable maps would succeed in recovering the rotated wavy distribution and instead fail for the standard one. This illustrates the subtle influence of input variable ordering on a map's approximation ability.

### 3.1.3 Basis functions

A very important part of the parameterization of the map component function $S_k$ is its construction: what basis functions should be used to represent $S_k$ and how much nonlinearity should they permit? In a nutshell,

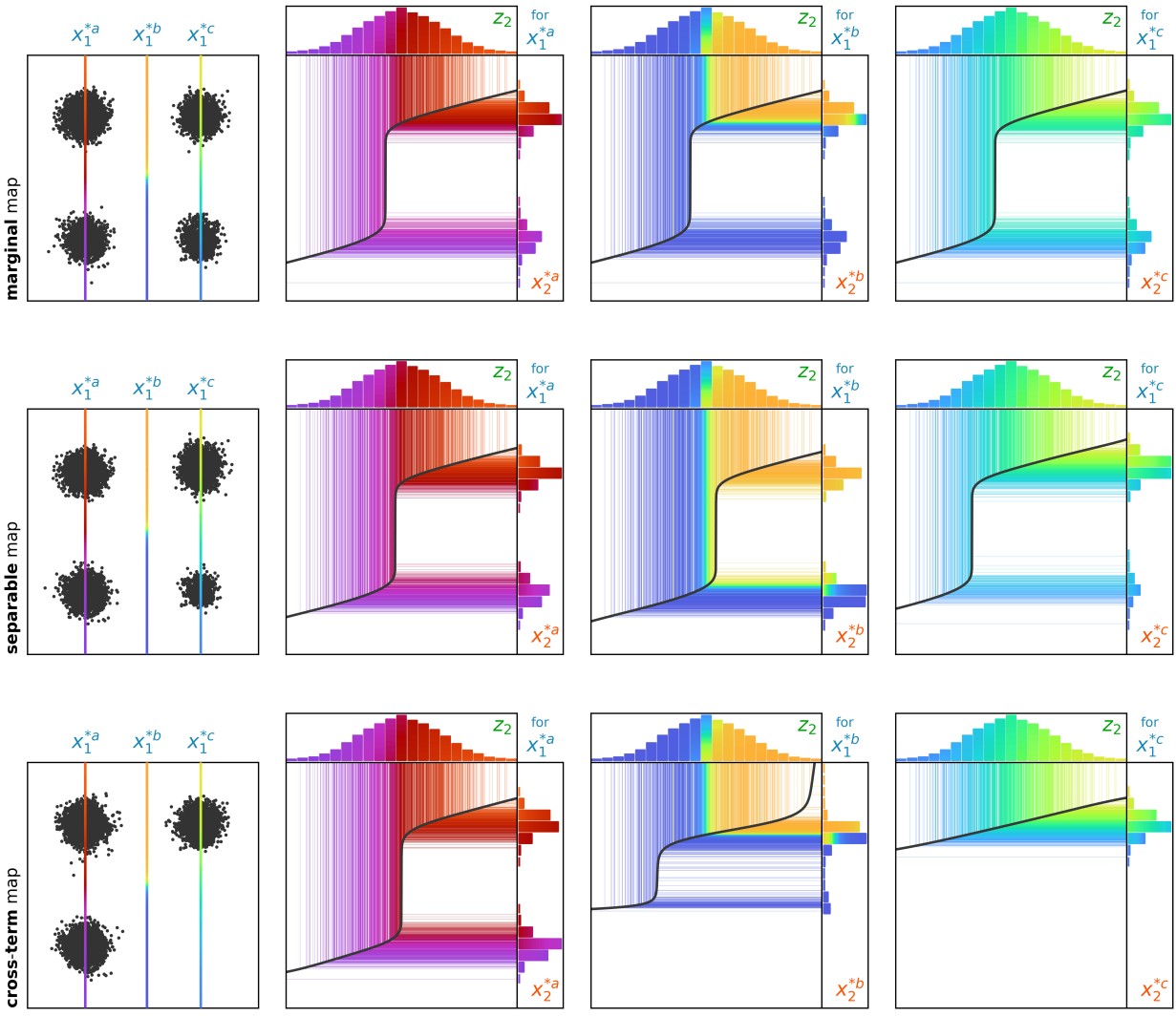

Figure 10: Conditional inversion $x_2 = S_2^{-1}(z_2; x_1)$ for marginal (top row), separable (center row), and cross-term (bottom row) maps. Each column corresponds to a conditional distribution for a different fixed $x_1^*$. Each term in Equation (25) has a different effect: $z_2$ defines the horizontal starting position for the inversion, $f(x_1, x_2)$ defines the shape of coupling, and $g(x_1)$ defines its horizontal offset. **Marginal** maps keep $f$ and $g$ fixed, and thus yield the same inverse $x_2$ for any previous $x_1$. **Separable** maps also keep $f$ constant, but can adjust the offset $g$. Finally, **cross-term** maps can adjust both $f$ and $g$ for different $x_1$, providing the greatest degree of flexibility.

simpler – perhaps separable – parameterizations are more computationally efficient, but can struggle to recover features of the target pdf $\pi$ that do not resemble the reference $\eta$, as discussed at the beginning of this section. By contrast, more complex parameterizations add the flexibility required to capture "localized" features of $\pi$ (e.g., skewness, multi-modality, ...), but risk unfavourable bias-variance trade-offs. While there are alternative choices (Lunde & Ramgraber, 2026), we discuss two common basis function families that trade off different advantages:

**Polynomial basis functions**   In function approximation, polynomials are a canonical choice for forming an approximation; in particular, harkening to polynomial chaos expansion and other traditional stochastic function approximation literature (Le Maître & Knio, 2010; Ernst et al., 2012), we can choose orthogonal polynomials for our approximation function class. Since this work focuses on unbounded $\mathbf{x}$, we will consider the probabilists' Hermite polynomial family $\{\mathrm{He}_j\}_{j=1}^{\infty}$, which has the orthogonality property

$$\int_{-\infty}^{\infty} \mathrm{He}_i(x)\mathrm{He}_j(x)\ \underbrace{\frac{1}{\sqrt{\tau}}\exp(-x^2/2)}_{\text{Gaussian weight}}\ dx = j!\delta_{ij} = \begin{cases} j! & j=i \\ 0 & j \neq i \end{cases} \tag{26}$$

where $\tau = 6.283185\ldots$ is twice Archimedes' constant. Broadly, Equation (26) shows what it means for polynomials of different orders (e.g., linear, quadratic, cubic, etc.) to be orthogonal under the Gaussian weight. Similar to orthogonal vectors in linear algebra, orthogonal polynomials (more generally, orthogonal functions) are often chosen as a basis for function approximation, as their elements contain no "overlapping information" with respect to a particular weight (which may or may not match $\eta$). As these polynomials are dependent on the pdf, a few other orthogonal polynomial families with their respective weights and support include Legendre polynomials (respecting a constant weight on $(-1, 1)$) and Laguerre polynomials (respecting weight $e^{-x}$ on $(0, \infty)$) (Szegő, 1939). These polynomial families may be suited for different problems depending on characteristics of $\eta$ and $\pi$ (Wang & Marzouk, 2022; Sharp et al., 2026).

Generally, if a polynomial family $\{\psi_j\}_{j=1}^{\infty}$ is orthogonal with respect to a weight $\rho$ (e.g., the probabilist Hermite polynomials take $\rho$ as the Gaussian weight), one can approximate a given well-behaved function $f(x)$ with $\hat{f}(x) = c_1\psi_1(x) + c_2\psi_2(x) + \cdots + c_P\psi_P(x)$ using a finite number $P$ of these polynomials. Guarantees for this approximation are often given when measuring the error according to our weight $\rho$ via $\int (f(x) - \hat{f}(x))^2\, d\rho(x)$ (Ernst et al., 2012). If we believe that $\eta$ and $\pi$ are very similar, choosing polynomials orthogonal with respect to our reference $\eta$ (which we know the orthogonal polynomial family of) will also perform well when the input is distributed according to $\pi$ (which we generally won't know the orthogonal polynomial family of). On the other hand, if $\eta$ and $\pi$ are not very similar, such guarantees are not particularly helpful; practitioners see this manifest as a need for more complex map parameterizations (see Section 3.1.2).

As they have unbounded support, Hermite polynomials exert global influence and often require high-order terms to capture fine distributional features. While versatile, the use of polynomial basis functions has a few pitfalls which demand caution:

1. **Importance of standardization**: In response to the above, we would like to make $\pi$ resemble $\eta$ more closely by pre-processing the data we have from $\pi$. When working with a Gaussian reference $\eta$, it is thus generally advised to standardize $\mathbf{x}$ by trying to match the mean and variance of $\eta$. We do this by (i) subtracting the empirical mean $\bar{\mathbf{X}} = 1/N\sum_{i=1}^{N}\mathbf{X}^i$ and (ii) scaling each marginal to unit variance by dividing it through the unbiased estimator of the empirical standard deviation $\sigma_k$ where $\sigma_k^2 = (N-1)^{-1}\sum_{i=1}^{N}(\mathbf{X}_k^i - \bar{\mathbf{X}}_k)^2$ before formulating and applying a transport map[10] $\mathbf{S}$. All subsequent map operations are then implemented in this standardized space. Finally, the map's outputs are transformed back into target space by reversing the standardization. In effect, standardization wraps a separate linear transformation around the transport map, and thereby does not affect the validity of the operations inside this wrapper. In the subsequent sections, we will assume by default that

---

[10] Alternatively, one may standardize the samples by forming the empirical cdf $\tilde{F}_{\pi_k}$ along the marginal for $x_k$, then standardize the samples by composing $\tilde{F}_{\pi_k}$ with the inverse standard Gaussian cdf $F_{\eta_k}^{-1}$, that is to say, $F_{\eta_k}^{-1} \circ \tilde{F}_{\pi_k}(x_k)$, ensuring each marginal of the standardized samples has a Gaussian distribution.

the samples have been standardized. As pointed out in Morrison et al. (2022), invertible diagonal transformations retain the conditional independence structure of the target distribution. In other words, there is no loss of information while working in the marginally rescaled space.

2. **Edge control**: A practical challenge of polynomials is their growth in the tails for higher-order terms. This often leads to volatility if the map is evaluated or inverted far from zero. To address this issue, one useful option is the use of *edge-controlled terms* such as *Hermite functions* $\mathcal{H}_j$. This variant of basis functions are defined as probabilist's Hermite polynomials $\mathrm{He}_j$ of order $j$ multiplied with a Gaussian weight, which reverts the polynomial's output to zero far from zero:

$$\mathcal{H}_j\left(x\right) = \mathrm{He}_j\left(x\right)\exp\left(-\frac{x^2}{4}\right). \tag{27}$$

A nice feature of this form is that $\mathcal{H}_i(x)\mathcal{H}_j(x) = \mathrm{He}_i(x)\mathrm{He}_j(x)\exp(-x^2/2)$, and so $\{\mathcal{H}_j\}_{j=1}^{\infty}$ will inherit the same error properties as $\{\mathrm{He}_j\}$ when measuring error without any weight $\rho$, i.e., $\int (f(x) - \hat{f}(x))^2\,dx$.

Illustrations of the first few orders of Hermite functions are provided in Figure 11A. Note that expressing the map exclusively in terms of Hermite functions causes the map component $S_k$ to revert to zero in the tails, which can be undesirable whenever extrapolation may be required. In practice, limiting the Gaussian weight term to polynomials of order two or larger is a good compromise, thus we recommend retaining the linear terms without a weight.

A practical issue with Hermite functions can emerge for target distributions $\pi$ which include sharp features. From (Szegő, 1939), the largest maximizer $x_j^*$ of $\mathrm{He}_j$ grows at a rate of $x_j^* = \sqrt{j} + \mathcal{O}(j^{-1/6})$, which means that if $\mathcal{H}_j$ has nontrivial values on $(-r, r)$, we will have that $\mathcal{H}_{4j}$ has nontrivial values on $(-2r, 2r)$; e.g., it is reasonable to estimate from Figure 11(A) that $r > 4$ for $j = 2$ and so $\mathcal{H}_8$ has nontrivial values on $(-8, 8)$. Due to this "global" nature of polynomials influencing the edges of the Hermite functions, maps for such distributions often involve high-order terms with high-magnitude coefficients of opposing signs partially cancelling each other. While this is often unproblematic within the support of the training samples, the resulting high-magnitude coefficients can extend the influence of some Hermite functions farther into the tails, resulting in undesirable tail effects. As an alternative, we might prefer to employ a weight term that reverts the weights to zero at a finite distance. For instance, using a cubic spline weight yields a different *edge-controlled* Hermite polynomial

$$\mathcal{H}_j^{\mathrm{EC}}\left(x\right) = \mathrm{He}_j\left(x\right)\left(2\min\left(1, |x|/r\right)^3 - 3\min\left(1, |x|/r\right)^2 + 1\right), \tag{28}$$

where $r$ is a finite edge-control radius specified by the user. Examples of such basis functions are illustrated in Figure 11B. When using these functions in the integrated-rectified formulation, this will create $\mathbf{S}$ that is linear outside of the hypercube $(-r, r)^d$.

**Radial basis functions**  A second useful class of basis functions is *radial basis functions* (RBFs). RBFs exert more local influence than combinations of polynomial basis functions, and generally require two additional parameters: a position parameter $\mu$ and a scale (or bandwidth) parameter $\sigma$. To avoid the need to optimize these parameters along with the map's coefficients $c_i$, one can estimate $\mu$ and $\sigma$ from the empirical quantiles along the marginals of the training samples $\{\mathbf{X}^i\}_{i=1}^{N}$ (Spantini et al., 2022). For the positions $\mu_i$, they propose to place each RBF at the empirical quantiles $q_{i/(j+1)}(x_k)$ for $i = 1, \ldots, j$, where $j$ is the total number of RBFs along the $k$-th marginal. Based on the resulting $\mu_i$, the corresponding scales $\sigma_i$ are determined by averaging the distances to neighbouring RBFs. To simplify notation, it can be useful to define local coordinates for each RBF:

$$x_k^{\mathrm{loc},i} = \frac{x_k - \mu_i}{\sqrt{\tau}\sigma_i}, \tag{29}$$

where the $\tau$ is again twice the Archimedes' constant. Radial basis functions are particularly useful for multimodal distributions because they render the map selectively expressive wherever the target distribution's

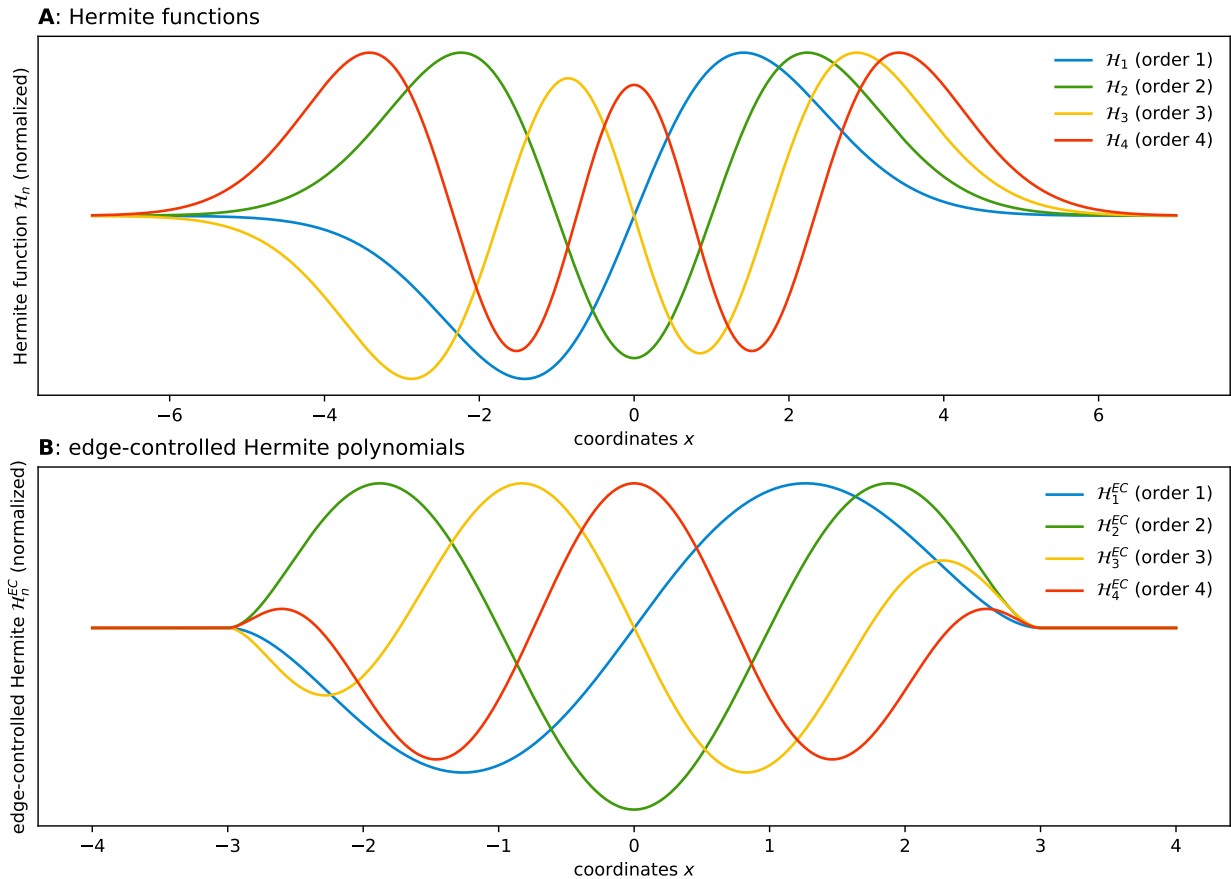

Figure 11: Two types of edge-controlled basis functions based on Hermite polynomials. (A) Hermite functions are multiplied with a Gaussian weight, causing them to approach zero in the tails. (B) Replacing the Gaussian weight with a scaled cubic spline leads to a different edge-controlled basis function. This enforces limited support, ensuring the function reverts to zero at a finite distance $r$.

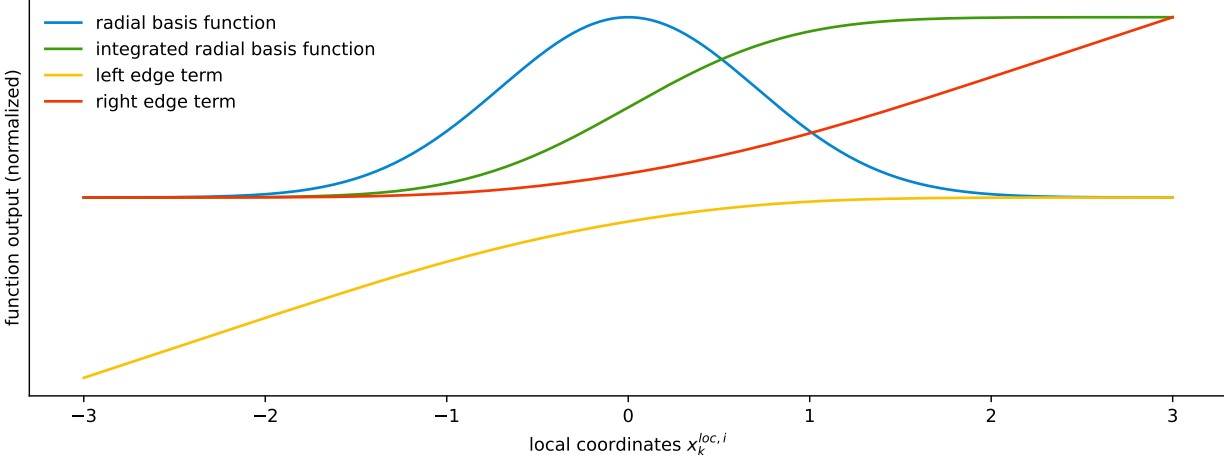

Figure 12: Illustration of different useful RBF basis functions.

samples are located. Where a superposition of many high-order polynomials may be required to recover isolated modes, often only a few RBFs may suffice. Besides conventional RBFs, three useful related basis functions are *left edge terms* (LET), *integrated RBFs* (iRBF), and *right edge terms* (RET):

$$
\begin{aligned}
\text{RBF}\left(x_k^{\text{loc},i}\right) &= \frac{1}{\sqrt{\tau}}\exp\left(-x_k^{\text{loc},i^2}\right), \\
\text{iRBF}\left(x_k^{\text{loc},i}\right) &= \frac{1}{2}\left(1+\text{erf}\left(x_k^{\text{loc},i}\right)\right), \\
\text{LET}\left(x_k^{\text{loc},i};\sigma_i\right) &= \frac{1}{2}\left(\sigma_i\sqrt{\tau}x_k^{\text{loc},i}\left(1-\text{erf}\left(x_k^{\text{loc},i}\right)\right)-\frac{4\sigma_i}{\sqrt{\tau}}\exp\left(-x_k^{\text{loc},i^2}\right)\right), \\
\text{RET}\left(x_k^{\text{loc},i};\sigma_i\right) &= \frac{1}{2}\left(\sigma_i\sqrt{\tau}x_k^{\text{loc},i}\left(1+\text{erf}\left(x_k^{\text{loc},i}\right)\right)+\frac{4\sigma_i}{\sqrt{\tau}}\exp\left(-x_k^{\text{loc},i^2}\right)\right).
\end{aligned}
\tag{30}
$$

Illustrations of these functions are provided in Figure 12. Note that iRBFs, LETs, and RETs are monotone functions, and thus an excellent choice for a *linear separable* map (see Section 3.1.2). Other sigmoid-like functions and monotone "edge terms" that revert to linear in the tails include logistic and softplus functions, which may be computationally faster.

## 3.2 Optimization

With the structure and parameterization of the map component functions $S_k$ defined, we may now address the question of how to identify the specific map $\mathbf{S}$ that relates the target distribution $\pi$ to the reference distribution $\eta$. In general, we have two different ways to estimate the map, depending on the type of information available: *maps from samples*, and *maps from densities*. Both follow a very similar approach, but differ fundamentally in the direction in which they define the transport map:

**Maps from samples**

- require target samples $\mathbf{x}\sim\pi$

- require evaluations of the reference $\eta$

- seek a map $\mathbf{S}$ such that $\mathbf{S}_\sharp\pi=\eta$

**Maps from densities**

- require reference samples $\mathbf{z}\sim\eta$

- require evaluations of unnormalized target $\tilde{\pi}$

- seek a map $\mathbf{R}$ such that $\mathbf{R}_\sharp\eta=\pi$

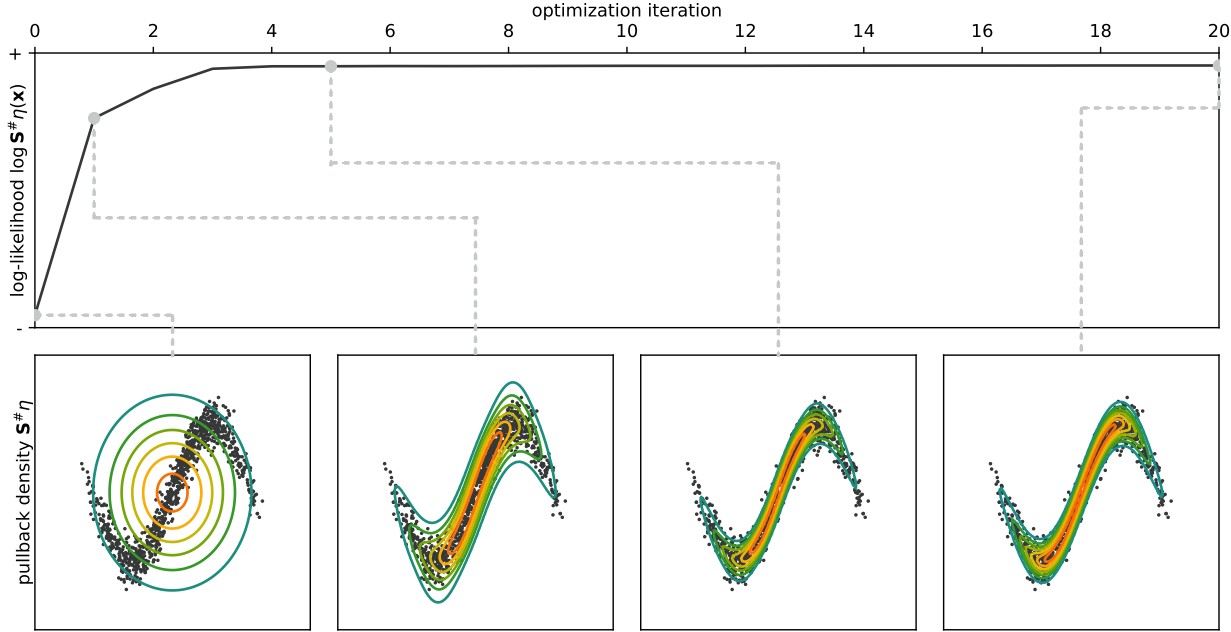

Figure 13: The optimization objective of *maps from samples* maximizes the log-likelihood of a set of fixed samples $\mathbf{x} \sim \pi$ over the map's pullback density $\mathbf{S}^{\sharp}\eta \approx \pi$. In practical terms, the optimization seeks maps $\mathbf{S}$ which mold the reference pdf $\eta$ to the target samples $\mathbf{x}$.

### 3.2.1 Maps from samples

In the preceding sections, we have often implicitly assumed a *maps from samples* scenario. In other words, the target pdf $\pi$ is only known through samples , i.e., we have a collection of samples $\mathbf{X}^i \sim \pi$. In this case, we seek to minimize the "distance" between the target pdf $\pi$ and the pullback pdf $\mathbf{S}^{\sharp}\eta$. While several notions of distance between pdfs are possible, the Kullback–Leibler divergence is commonly used for its computational tractability. For two arbitrary pdfs $p, q$ and a generic RV $\mathbf{a}$, we define their Kullback–Leibler divergence $D(p\|q)$ as:

$$D(p\|q) = \mathbb{E}_p \left[ \log \frac{p(\mathbf{a})}{q(\mathbf{a})} \right] = \int p(\mathbf{a}) \log \frac{p(\mathbf{a})}{q(\mathbf{a})} d\mathbf{a} \tag{31}$$

Thus, we can formulate our optimization problem as finding the map $\mathbf{S}_{\mathrm{opt}}$ that minimizes the difference, measured in terms of the Kullback–Leibler divergence, between the target $\pi$ and its approximation $\mathbf{S}^{\sharp}\eta \approx \pi$:

$$\mathbf{S}_{\mathrm{opt}} = \arg \min_{\mathbf{S} \in \mathcal{F}} D(\pi \| \mathbf{S}^{\sharp}\eta), \tag{32}$$

where $\mathcal{F}$ is some appropriate class of functions (see Section 3.1). This is often referred to as the minimization of the *forward* Kullback–Leibler divergence. We note that $D(\pi\|\mathbf{S}^{\sharp}\eta)$ can be expanded as:

$$D(\pi\|\mathbf{S}^{\sharp}\eta) = \mathbb{E}_\pi \left[ \log \frac{\pi(\mathbf{x})}{\mathbf{S}^{\sharp}\eta(\mathbf{x})} \right] = \mathbb{E}_\pi \left[ \log \pi(\mathbf{x}) \right] - \mathbb{E}_\pi \left[ \log \mathbf{S}^{\sharp}\eta(\mathbf{x}) \right], \tag{33}$$

where the first term $\mathbb{E}_\pi \left[ \log \pi(\mathbf{x}) \right]$ does not depend on the map $\mathbf{S}$. Thus, we can alternatively view Equation (32) as an attempt to maximize the log-likelihood (or minimize the negative log-likelihood) of the map's pullback density $\mathbf{S}^{\sharp}\eta \approx \pi$ over the target density $\pi$:

$$\mathbf{S}_{\mathrm{opt}} = \arg \min_{\mathbf{S} \in \mathcal{F}} \mathbb{E}_\pi \left[ - \log \mathbf{S}^{\sharp}\eta(\mathbf{x}) \right]. \tag{34}$$

An illustration of the optimization process is provided in Figure 13. Substituting in Equation (4) for $\mathbf{S}^\sharp \eta(\mathbf{x})$ into the cost function Equation (34) and expanding the logarithms yields:

$$\mathbb{E}_\pi \left[ -\log \mathbf{S}^\sharp \eta(\mathbf{x}) \right] = \mathbb{E}_\pi \left[ -\log \eta(\mathbf{S}(\mathbf{x})) - \log |\det \nabla_\mathbf{x} \mathbf{S}(\mathbf{x})| \right]. \tag{35}$$

Due to the lower triangular structure of $\mathbf{S}$, we know that $\nabla_\mathbf{x} \mathbf{S}$ is a lower triangular matrix. As established in Equation (6), the determinant of a lower triangular matrix is given by the product of its diagonal entries:

$$\log \det \nabla_\mathbf{x} \mathbf{S}(\mathbf{x}) = \log \left[ \prod_{k=1}^{K} \frac{\partial S_k(\mathbf{x})}{\partial x_k} \right] = \sum_{k=1}^{K} \log \frac{\partial S_k(\mathbf{x})}{\partial x_k}. \tag{36}$$

Mind that due to the monotonicity of $S_k$ in $x_k$ (see Section 3.1.1), the derivative $\partial_{x_k} S_k$ will always be positive, and the logarithm in the right-hand side sum of Equation (36) will be defined. Plugging this identity into the cost function, we only now enforce that our reference $\eta$ is multivariate Gaussian by substituting its pdf into the expectation.

$$\mathbb{E}_\pi \left[ -\log \mathbf{S}^\sharp \eta(\mathbf{x}) \right] = \mathbb{E}_\pi \left[ -\log \frac{1}{\tau^{D/2}} + \frac{1}{2} \sum_{k=1}^{K} S_k(\mathbf{x})^2 - \sum_{k=1}^{K} \log \frac{\partial S_k(\mathbf{x})}{\partial x_k} \right] \tag{37}$$

where $\tau$ is again twice the Archimedes' constant. Recognizing that the first term is constant with respect to $\mathbf{S}$, it can be discarded from the objective function. Using $N$ samples from the target distribution $\mathbf{X}^i \sim \pi$ and merging the two sums over $K$, we obtain a Monte Carlo approximation of the loss function:

$$\mathcal{J}(\mathbf{S}) = \sum_{i=1}^{N} \sum_{k=1}^{K} \left( \frac{1}{2} S_k(\mathbf{X}^i)^2 - \log \frac{\partial S_k(\mathbf{X}^i)}{\partial x_k} \right). \tag{38}$$

From this expression, we can recognize that the summands of the full objective function $\mathcal{J}(\mathbf{S})$ are independent of each other. As a consequence, we can define separate objective functions $\mathcal{J}_k(S_k)$ for each map component $S_k$:

$$\mathcal{J}_k(S_k) = \sum_{i=1}^{N} \left( \frac{1}{2} S_k(\mathbf{X}^i)^2 - \log \frac{\partial S_k(\mathbf{X}^i)}{\partial x_k} \right). \tag{39}$$

Equation (39) may then be minimized with off-the-shelf software developed for other optimization and machine learning tasks. If the linear separable formulation in Section 3.1.1 is chosen, it is only necessary to optimize the monotone part $f$ of each map component $S_k$ numerically, as the coefficients for the nonmonotone terms $g$ can be derived in closed form. More detail on this is provided in Section A. Note that since the objective functions in Equation (39) are independent of each other, the map component functions $S_k$ can be optimized in arbitrary order, even in parallel. Each objective function $\mathcal{J}_k$ balances a first quadratic term that drives samples $S_k(\mathbf{X}^i)^2$ to zero with a second term that acts as a log-barrier function to prevent null diagonal entries in the gradient of the transport map (Le Provost et al., 2021).

### 3.2.2 Maps from densities

Alternatively, we choose a different optimization strategy in the case that we can evaluate the target pdf $\pi$, but only up to a constant of proportionality; i.e., we have access to evaluating density $\tilde{\pi} = m\pi$ for some unknown $m > 0$. A key difference when learning a *map from density* is that we choose to define the triangular map in the opposite direction as before: where we previously considered a forward map $\mathbf{S}$ that maps target samples $\mathbf{X}$ to reference samples $\mathbf{Z}$, we now define a forward map $\mathbf{R}$ mapping reference samples $\mathbf{Z}$ to target samples $\mathbf{X}$. Correspondingly, the pushforward distribution $\mathbf{R}_\sharp \eta$ now approximates the target distribution $\pi$ via the forward map $\mathbf{R}$, and the pullback distribution $\mathbf{R}^\sharp \pi$ now approximates the reference distribution $\eta$ via the inverse map $\mathbf{R}^{-1}$. To find the optimal $\mathbf{R}_{\mathrm{opt}}$, we seek to minimize the Kullback–Leibler divergence between the reference distribution $\eta$ and the pullback distribution $\mathbf{R}^\sharp \pi$ (Marzouk et al., 2017):

$$\mathbf{R}_{\mathrm{opt}} \in \arg \min_{\mathbf{R} \in \mathcal{F}} D(\eta \| \mathbf{R}^\sharp \pi), \tag{40}$$

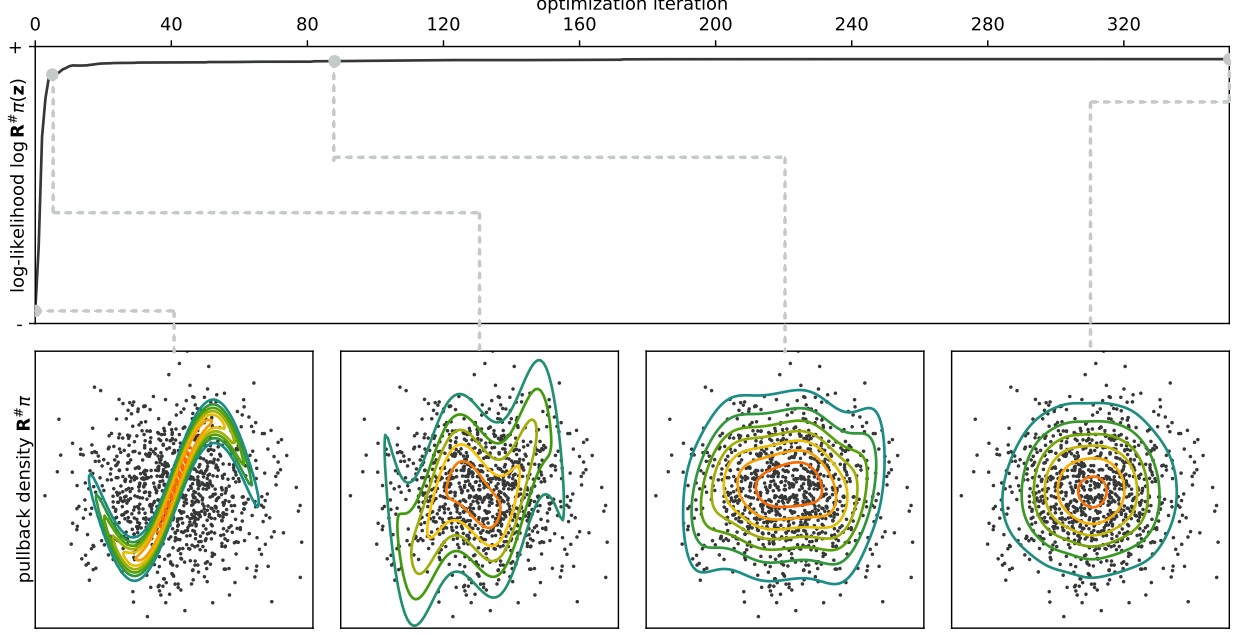

Figure 14: The optimization objective of *maps from densities* maximizes the log-likelihood of a set of fixed samples $\mathbf{Z} \sim \eta$ over the map's pullback density $\mathbf{R}^{\sharp}\tilde{\pi} \approx \eta$. In practical terms, the optimization seeks maps $\mathbf{R}$ which mold the unnormalized target pdf $\tilde{\pi}$ to the reference samples $\mathbf{Z}$.

where $\mathcal{F}$ is some appropriate class of functions (see Section 3.1.3) and $\in$ is used to suggest multiple minimizers (see Baptista et al. (2023) for a discussion of why this could happen). This is often referred to as the minimization of the *reverse* Kullback–Leibler divergence. An illustration of the minimization in Equation (40) is provided in Figure 14. Similar to Equation (33) in the maps from samples scenario, we expand $D(\eta \| \mathbf{R}^{\sharp}\pi)$ as:

$$D(\eta \| \mathbf{R}^{\sharp}\pi) = \mathbb{E}_{\eta}\left[\log \frac{\eta(\mathbf{z})}{\mathbf{R}^{\sharp}\pi(\mathbf{z})}\right] = \mathbb{E}_{\eta}\left[\log \eta(\mathbf{z})\right] - \mathbb{E}_{\eta}\left[\log \mathbf{R}^{\sharp}\pi(\mathbf{z})\right], \tag{41}$$

where the first term $\mathbb{E}_{\eta}\left[\log \eta(\mathbf{z})\right]$ does not depend on the map $\mathbf{R}$. Thus, Equation (40) can be viewed as an attempt to maximize the log-likelihood (or minimize the negative log-likelihood) of the map's pullback density $\mathbf{R}^{\sharp}\pi \approx \eta$ over the reference density $\eta$:

$$\mathbf{R}_{\text{opt}} \in \arg \min_{\mathbf{R} \in \mathcal{F}} \mathbb{E}_{\eta}\left[-\log \mathbf{R}^{\sharp}\pi(\mathbf{z})\right]. \tag{42}$$

We may now substitute the pullback density from the change-of-variables formula (Equation (4)):

$$\eta(\mathbf{z}) \approx \mathbf{R}^{\sharp}\pi(\mathbf{z}) = \pi(\mathbf{R}(\mathbf{z})) \left|\det \nabla_{\mathbf{z}} \mathbf{R}(\mathbf{z})\right|. \tag{43}$$

Equivalent to Equation (36), we express the log determinant of the forward map $\mathbf{R}$ as a product of its diagonal entries:

$$\log \det \nabla_{\mathbf{z}} \mathbf{R}(\mathbf{z}) = \log \left[\prod_{k=1}^{K} \frac{\partial R_k(\mathbf{z})}{\partial z_k}\right] = \sum_{k=1}^{K} \log \frac{\partial R_k(\mathbf{z})}{\partial z_k}. \tag{44}$$

Substituting the two equations above into the cost function of Equation (42), we obtain:

$$\mathbb{E}_{\eta}\left[-\log \mathbf{R}^{\sharp}\pi(\mathbf{z})\right] = -\mathbb{E}_{\eta}\left[\log \pi(\mathbf{R}(\mathbf{z}))\right] - \mathbb{E}_{\eta}\left[\sum_{k=1}^{K} \log \frac{\partial R_k(\mathbf{z})}{\partial z_k}\right]. \tag{45}$$

Since we assume we only have access to an unnormalized target density $\tilde{\pi}$, we can expand the first term $\mathbb{E}_\eta \left[ \log \pi(\mathbf{R}(\mathbf{z})) \right]$ using the identity $\pi = \tilde{\pi}/m$, where $m$ is the unknown normalization factor:

$$\mathbb{E}_\eta \left[ -\log \mathbf{R}^\sharp \pi(\mathbf{z}) \right] = -\mathbb{E}_\eta \left[ \log \tilde{\pi}(\mathbf{R}(\mathbf{z})) \right] + \mathbb{E}_\eta \left[ \log m \right] - \mathbb{E}_\eta \left[ \sum_{k=1}^{K} \log \frac{\partial R_k(\mathbf{z})}{\partial z_k} \right]. \tag{46}$$

Recognizing that the normalization factor $m$ does not depend on the map $\mathbf{R}$, we can discard it from the objective function:

$$\mathbf{R}_{\text{opt}} \in \arg\min_{\mathbf{R} \in \mathcal{F}} -\mathbb{E}_\eta \left[ \log \tilde{\pi}(\mathbf{R}(\mathbf{z})) \right] - \mathbb{E}_\eta \left[ \sum_{k=1}^{K} \log \frac{\partial R_k(\mathbf{z})}{\partial z_k} \right]. \tag{47}$$

Using $N$ samples from the reference distribution $\mathbf{Z}^i \sim \eta$, we obtain a Monte Carlo approximation of the loss function:

$$\mathcal{J}(\mathbf{R}) = \sum_{i=1}^{N} \left( -\log \tilde{\pi} \left( \mathbf{R} \left( \mathbf{Z}^i \right) \right) - \sum_{k=1}^{K} \log \frac{\partial R_k(\mathbf{Z}^i)}{\partial z_k} \right). \tag{48}$$

This yields the objective function for optimizing maps from densities. Two comments are in order regarding this optimization problem. First, observe that opposed to maps from samples, we have to first generate an ensemble of $N$ i.i.d. samples $\mathbf{Z}^i \sim \eta$. Second, since we now find a map $\mathbf{R}$ to pulls the reference back to the target, we still avoid evaluating the inverse map $\mathbf{R}^{-1}$ when calculating the loss $\mathcal{J}(\mathbf{R})$ for some candidate map $\mathbf{R} \in \mathcal{F}$. However, opposed to optimizing a map from samples (Section 3.2.1), this objective function cannot generally be subdivided into individual objectives for each constituent map component $R_k$. As such, all map components are usually found via a single optimization problem.

We note that there are other loss functions that we could consider beyond the Kullback–Leibler divergence, for instance the variance diagnostic (El Moselhy & Marzouk, 2012; Richter et al., 2020).We do not consider these options here in greater detail, as the Kullback–Leibler objective is widely used and quite tractable. Moreover, in the maps-from-samples setting, a minimizer of the forward Kullback–Leibler divergence corresponds to a *maximum likelihood estimator* of the transport map within the chosen function class; this link is useful for both interpretation and theory (Wang & Marzouk, 2022).

### 3.2.3 Regularization

When the number of samples $N$ is small relative to either the target's dimensionality $K$ or the number of parameters for the map components, using more complex maps (i.e., increasing the number of the parameters) risks overfitting. In turn, this might lead to numerical instabilities. To prevent these issues, we can introduce an $L^1$ or $L^2$ regularization penalty to the objective functions introduced in the preceding section. In the maps from samples (Section 3.2.1) case, we take map component $S_k$ for some fixed $k$, and let $c_i$ parameterize this component via, e.g., polynomial coefficients. The regularized objective function would be:

$$\mathcal{J}_k^{\text{L1}}(S_k) = \sum_{i=1}^{N} \left( \frac{1}{2} S_k(\mathbf{X}^i)^2 - \log \frac{\partial S_k(\mathbf{X}^i)}{\partial x_k} \right) + \sum_{i=1}^{j} \lambda_i |c_i| \quad L^1 \text{ regularization,}$$

$$\mathcal{J}_k^{\text{L2}}(S_k) = \sum_{i=1}^{N} \left( \frac{1}{2} S_k(\mathbf{X}^i)^2 - \log \frac{\partial S_k(\mathbf{X}^i)}{\partial x_k} \right) + \sum_{i=1}^{j} \lambda_i c_i^2 \quad L^2 \text{ regularization.}$$

(49)

where we add a $\lambda$-weighted penalty term to the objective functions, which penalizes non-zero parameters $c_i$ for each of the $i = 1, \ldots, P_k$ basis functions of map component $S_k$ (see Section 3.1.3). This creates an optimum closer to where the parameters are zero, which can be interpreted as penalizing map complexity. The regularization factors $\lambda_i$ are user-specified hyperparameters that should be tuned to the problem at hand, ensuring that the penalization is large enough to have an effect and small enough to avoid excessive biasing. For maps from densities, the optimization objective follows an equivalent structure to Equation (49).

Similar to other applications with regularization parameters, a bit of experimentation is required to find a suitable balance between regularization and bias.

In practice, the use of $L^1$ regularization is a common strategy to discover sparsity in the triangular map, which we recall corresponds to conditional independence (see Section 2.3.3). In many cases, it is more efficient to impose sparsity and parsimony by construction. We will discuss practical strategies to learn a parsimonious degree of map complexity in Section 4.2.

## 4 Practical heuristics

With the theoretical foundations and implementation-related details established, we now discuss a few important features and tricks that help to further enhance the potential of triangular transport maps.

### 4.1 Composite maps for conditional sampling

For complex target distributions $\pi$, it is often infeasible to define and optimize a map of sufficient complexity to capture all features of $\pi$. In such cases, the map $\mathbf{S}$ will fail to completely normalize the target $\pi$, making certain features persist in the pushforward $\mathbf{S}_\sharp \pi$. At the same time, the pullback $\mathbf{S}^\sharp \eta$ may not be a good approximation of $\pi$. Figure 15 illustrates this for two different levels of map complexity.

This imperfection has important consequences for Bayesian inference with triangular transport maps. Since the (conventional) map's conditional inverse only samples a conditional of the pullback distribution $\mathbf{S}^\sharp \eta$, not of the real target $\pi$, the quality of the resulting posterior samples will depend on the mismatch between $\mathbf{S}^\sharp \eta$ and $\pi$. Yet, we are often given a sample $\mathbf{X} = (\mathbf{X}_{1:k}, \mathbf{X}_{k+1:K})$ of a joint distribution $\pi(\mathbf{x}_{1:k}, \mathbf{x}_{k+1:K})$ and would like to create a high-quality sample $\mathbf{X}^*_{k+1:K}$ conditioned on some fixed $\mathbf{x}^*_{1:k}$. To do this, we apply a *composite map* to each sample $\mathbf{x}$, defined as

$$\mathbf{x}^*_{k+1:K} = \mathbf{S}^{-1}_{k+1:K}(\cdot \; ; \; \mathbf{x}^*_{1:k}) \circ \mathbf{S}_{k+1:K}(\mathbf{x}_{k+1:K}; \mathbf{x}_{1:k}). \tag{50}$$

Note that $\mathbf{x}^*_{1:k}$ and $\mathbf{x}_{1:k}$ are two different objects: The former comes from the real world (i.e., it is a fixed value, independent from the random sample $\mathbf{x}$). For instance, $\mathbf{x}^*_{1:k}$ may be specific measurement values on which we want to condition $\pi$. By contrast, the latter are jointly sampled with $\mathbf{x}_{k+1:K}$ from $\pi$. While this may initially seem complicated, the key idea of composite maps is as follows:

1. We have an approximate map $\mathbf{S}$. Its **forward evaluation** thus does not transform samples from the true target $\mathbf{x} \sim \pi$ into samples from the reference $\mathbf{z} \sim \eta$, but instead yields samples from the pushforward $\tilde{\mathbf{z}} \sim \mathbf{S}_\sharp \pi \neq \eta$. These pushforward samples $\tilde{\mathbf{z}}$ preserve features of the target distribution $\pi$ the map did not capture. For instance, a linear map $\mathbf{S}$ can only shift, scale, and rotate the target distribution $\pi$, which is insufficient if $\pi$ is non-Gaussian (Figure 15A).

2. Likewise, an approximate **inverse map** does not transform samples from the reference $\mathbf{z} \sim \eta$ into samples from the true target $\mathbf{x} \sim \pi$, instead yielding samples from the pullback $\tilde{\mathbf{x}} \sim \mathbf{S}^\sharp \eta \neq \pi$ which lack any features of $\pi$ that the map has not captured. For instance, the pullback $\mathbf{S}^\sharp \eta$ of a linear map $\mathbf{S}$ only samples a Gaussian approximation of the target distribution (Figure 15B).

3. Applying an invertible approximate forward map, directly followed by its inverse, cancels the map's approximation error and always restores the original target samples. By construction, we know that the composition of $\mathbf{S}$ with its inverse $\mathbf{S}^{-1}$ maps a sample to itself, i.e., if $\tilde{\mathbf{z}} = \mathbf{S}(\mathbf{x}) \sim \mathbf{S}_\sharp \pi$, then $\mathbf{x} = \mathbf{S}^{-1}(\tilde{\mathbf{z}})$.

4. This restoration of unresolved features persists in part during the conditioning operation. Instead of reference samples $\mathbf{z} \sim \eta$ (Figure 16B), we might thus use pushforward samples $\tilde{\mathbf{z}} \sim \mathbf{S}_\sharp \pi$ to preserve some features of $\pi$. This generally reduces the map's approximation error (Figure 16C); see Proposition 11 in Baptista (2022).

Figure 16 illustrates the composite map's preservation of uncaptured features during conditional inversion for a Gaussian mixture target $\pi$ (Figure 16A). The pullback approximation $\mathbf{S}^\sharp \eta$ from a linear transport map

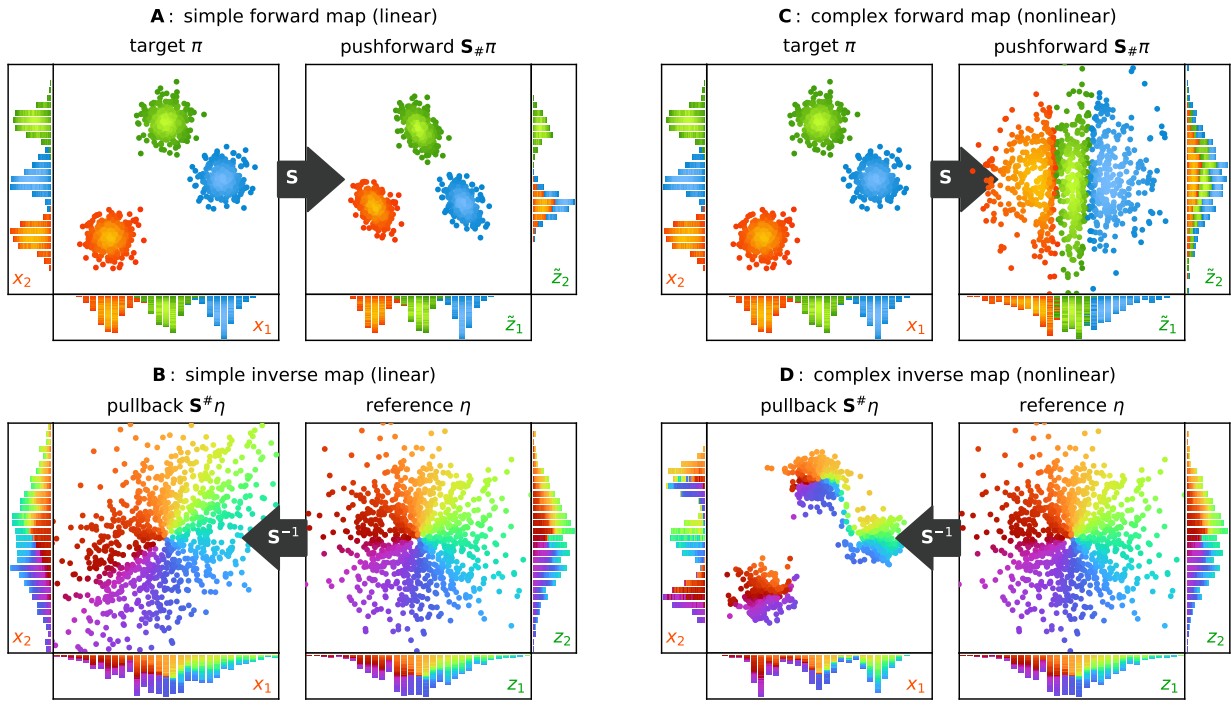

Figure 15: Simple maps capture only simple features. (A) For a non-Gaussian target $\pi$, a simple linear map does not yield a Gaussian pushforward $\mathbf{S}_\sharp \pi \neq \eta$. (B) Likewise, its pullback $\mathbf{S}^\sharp \eta \neq \pi$ does not provide a good approximation to the target. (C & D) More complex nonlinear maps yield better approximations. Color in (A) and (C) indicates the sample's original cluster membership, whereas in (B) and (D) color indicates a sample position's angle w.r.t. $\eta$'s mean.

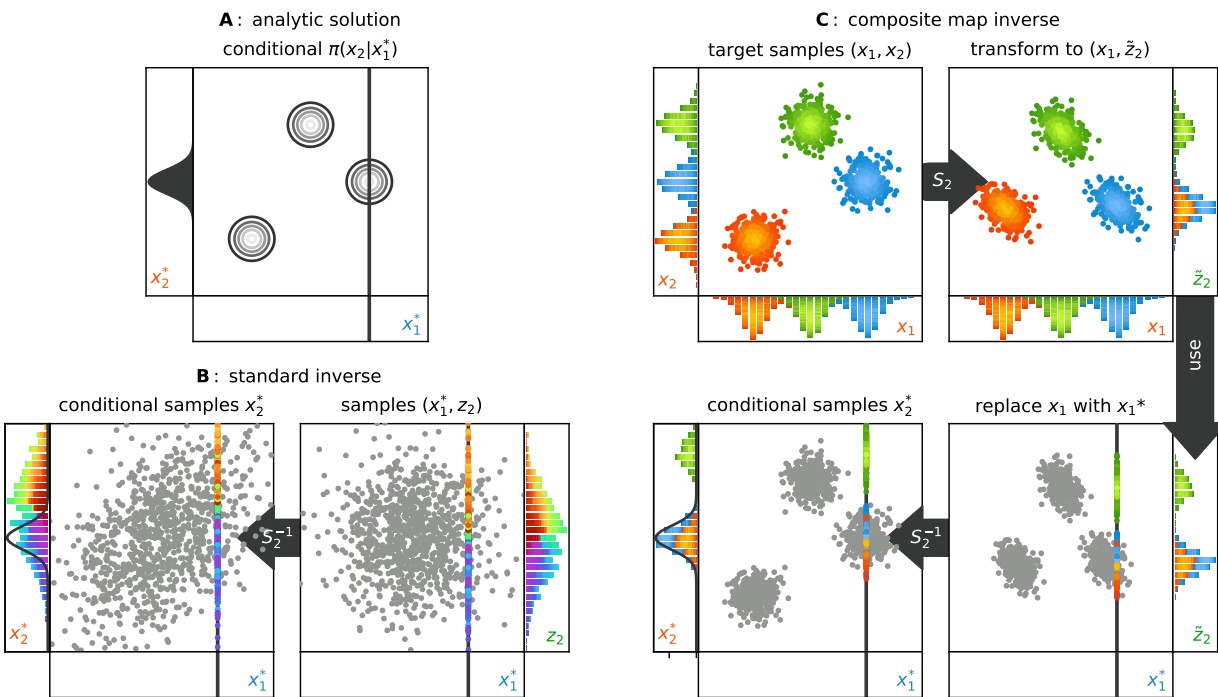

Figure 16: Composite maps yield better conditionals for imperfect maps. (A) The true solution for the conditioning operation. For the simple linear map, (B) using reference samples $z \sim \eta$ directly samples a conditional of the pullback $\mathbf{S}^\sharp \eta \neq \pi$, which here is unrepresentative of the true conditional. (C) Using pushforward samples $\tilde{z} \sim \mathbf{S}^\sharp \eta$ instead as part of a composite inversion yields a better approximation to the true conditional. The black line in the $x_2^*$ histograms in B, C, and D shows the true conditional distribution from subplot A for comparison.

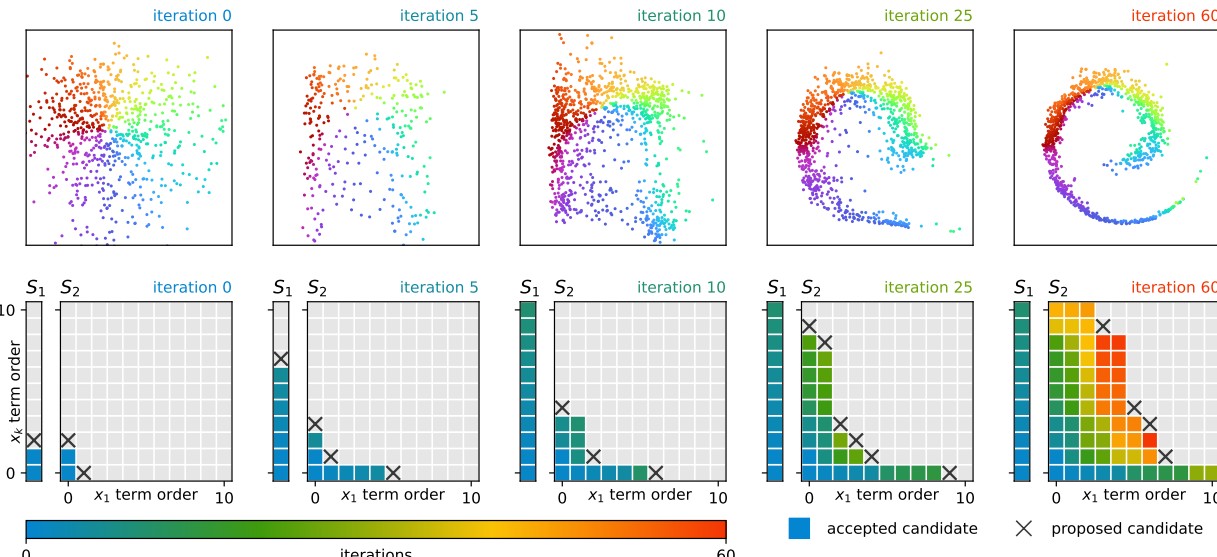

Figure 17: Adaptive transport maps gradually expand the complexity and expressiveness of the map components, adding terms corresponding to the largest absolute gradient of the objective function (Equation (39)) among a set of candidate terms. This identifies the necessary degree of map complexity gradually. The lower-row grids illustrate the order and composition of the map components at each iteration. Each filled cell is an accepted candidate, with coordinates $(3, 5)$ in the $S_2$ block corresponding to a term $\mathcal{H}_3(x_1)\mathcal{H}_5(x_2)$.

is Gaussian, and thus clearly inadequate for a multimodal Gaussian mixture target $\pi$. Inverting the map with reference samples $z \sim \eta$ only samples conditionals of the (multivariate Gaussian) pullback (Figure 16B). However, using reference samples from the pushforward $z \sim \mathbf{S}_\sharp \pi$ preserves complex features of the target $\pi$, yielding better conditional samples $x_2^*$ (Figure 16C).

Keep in mind that if we instead use a sufficiently complex, nonlinear map formulation (e.g., Figure 15D), the map approximation retrieves the target distribution sufficiently well. In this case, there is little practical difference between the use of reference $\mathbf{z} \sim \eta$ or pushforward samples $\tilde{\mathbf{z}} \sim \mathbf{S}_\sharp \pi$, as both distributions will be quite similar ($\mathbf{S}_\sharp \pi \approx \eta$). In the general case, however, we may not be certain about the quality of our map approximation. In consequence, we recommend the use of composite maps as a safer strategy for conditional sampling.

## 4.2  Map adaptation

Identifying a suitable degree of map complexity can be a challenging task. For one, conditional independence structures (see Section 2.3.3) are not always known a-priori. Similarly, finding a suitable degree of complexity for each map component to achieve an optimal bias-variance trade-off is not an easy task. To address these issues, we may draw on map adaptation algorithms. An example of such an algorithm is provided by Baptista et al. (2023).

In brief, the method starts with a triangular map that is viable and particularly simple; for example, one may use the identity map $S_k(x_k) = x_k$. The algorithm then proposes a list of candidate basis functions for each $S_k$ (Figure 17). These candidate terms either (i) extend the dependence of $S_k$ to a previous state dimension $x_{1:k-1}$, or (ii) incrementally increase the complexity of an existing dependence by adding higher degrees of nonlinearity.

To determine which candidate basis functions are added to the map, the algorithm calculates the gradient of the optimization objective function (see Section 3.2) with respect to the candidates' coefficients. It then adds the candidate corresponding to the steepest derivative. The idea behind this approach is that candidate terms with steep optimization objective gradients promise to improve the map's approximation significantly,

whereas those with flat gradients add little beyond complexity. The algorithm then proposes new candidates to the expanded map further and repeats the procedure until a user defined stopping criterion is met.

This process gradually expands the complexity and expressiveness of each map component. Since this process is iterative, this adaptation algorithm can be expensive if the number of candidate basis functions is large, as is the case for $S_k$ with many variable dependencies, particularly if the map parameterization permits cross-terms (Figure 17). The use of maps without cross-terms can drastically reduce this computational demand, as it would restrict the expansion and exploration of candidates to basis functions corresponding to the outer edges in Figure 17, making a two-dimensional result look like the letter "L".

## 5    Summary & Outlook

### 5.1    State of the art

Measure transport methods comprise an active and quickly evolving field in statistics and machine learning. Within this broader field, triangular maps occupy an important niche: transparent and versatile, they provide a powerful set of tools for conditional sampling from limited information. This, in turn, makes them highly useful for Bayesian inference.

To summarize, triangular transport methods have a number of important **advantages**:

1. **Parsimony**: The ability to fine-tune the parameterization of a triangular map, coupled with the sparse variable dependence that a triangular map inherits from conditional independence, allows triangular transport to be adapted to the computational demands, data availability, and distributional complexity of a given problem – increasing efficiency and accuracy.

2. **Numerical convenience**: Triangular maps are simple to learn and invert, and many common parameterizations provide optimization guarantees for the problem of learning maps from samples (see, e.g., Section A and Baptista et al. (2023)).

3. **Transparency**: Triangular maps provide a close correspondence between the map component functions $S_k$ and a specific factorization of the target distribution $\pi$. This link makes it easy to predict the impact of changes to the map on the resulting statistical model, and often allows us to decompose an inference or sampling task into smaller problems that are themselves interpretable.

At the same time, triangular maps face some important challenges. Ensuring the scalability of triangular transport with the dimension of the target distribution demands exploiting conditional independence to produce a sparse map, which in turn requires an appropriate variable ordering. We reiterate that, under mild assumptions, a triangular map exists between any target and reference (Santambrogio, 2015, Section 2.3), which includes any permutation of our target random variables. In practice, however, these maps may vary in sparsity and other notions of complexity: see Section 2.3.3 and Figure 9, respectively, and note that the rotated "wavy" distribution in Figure 9 actually corresponds to a different variable ordering. There are many approaches to finding orderings that maximize sparsity: for example, minimum degree algorithms (e.g., Amestoy et al., 1996; Cuthill & McKee, 1969) from graph theory, or other algorithms inspired by sparse Cholesky factorizations (Baptista et al., 2024d; Schäfer et al., 2021). Setting aside sparsity, however, the impact of variable ordering on the difficulty of *approximating* a given map component function $S_k$ is generally more difficult to predict, as it depends on finer properties of the target distribution at hand.

### 5.2    Broader context within transport

There are many ways to parameterize a triangular transport map that are not discussed here – for example, performing a closed-form integration over a parameterization of the target pdf. This can be done using squared polynomials (Zanger et al., 2024), parameterizing the square-root of the pdf (Dolgov et al., 2020), or even a composition of polynomials parameterizing this square-root pdf (Cui & Dolgov, 2022), often in the context of numerical tensor methods that permit efficient integration. Regarding the *approximation* of

triangular transport maps, there are several works elucidating approximation rates by, e.g., polynomials or neural networks (Zech & Marzouk, 2022a;b; Baptista et al., 2024b; Westermann & Zech, 2023); other works analyse the *statistical* consistency and convergence of triangular maps learned from finite samples (Wang & Marzouk, 2022; Irons et al., 2022).

More generally, triangular maps sit within the broader field of measure transport, which has seen remarkable computational advances in recent years. Many other approaches to constructing transport maps exchange triangular structure for different assumptions or desiderata. For example, *optimal* transport methods seek a mapping between two distributions that is as "close" to the identity function as possible, where closeness is encoded by a particular integrated *transport cost* (for instance, the distance between the input and output of the map) (Peyré et al., 2019). Many works seek to approximate optimal transport maps by searching over an appropriate function class – e.g., by parameterizing classes of convex functions (rather than our monotone triangular formulation) and writing the map as the gradient of such a function (Makkuva et al., 2020; Wang et al., 2023). There are some connections between triangular maps and optimal transport maps. First, each component $S_k$ of a monotone triangular map is optimal in its last (scalar) input $x_k$, conditioned on the first $k-1$ inputs $x_{1:k-1}$. *Conditional* optimal transport maps (Tabak et al., 2021; Carlier et al., 2016; Baptista et al., 2024c; Pooladian et al., 2025) generalize this idea from strictly triangular to block triangular structure. As described in Carlier et al. (2009), triangular maps also arise as the limit of optimal transport maps obtained with increasingly anisotropic quadratic transport cost.

Normalizing flows (Papamakarios et al., 2021) are another widely used class of transport methods, which typically do not seek to approximate some canonical (i.e., optimal or triangular) map, but rather use a composition of many simpler invertible transformations to construct a transport map, increasing the number of functions (layers) in the composition until the desired expressivity is reached. Autoregressive normalizing flows in fact use triangular maps as a building block, with parameterizations ranging in complexity from relatively simple to the complex rectified formulations discussed in Section 3.1.1 (Wehenkel & Louppe, 2019; Jaini et al., 2019). These flows interleave such triangular layers with permutations of the variables, which mitigates ordering issues described above at the cost of sacrificing the conditional independence and sparsity properties of triangular maps. Such constructions are partly inspired by earlier autoregressive models that directly parameterize marginal conditional densities (Bond-Taylor et al., 2021). Finally, many contemporary flow-based generative models (Song et al., 2020; Albergo et al., 2023; Lipman et al., 2023; Liu et al., 2023) can be understood as dynamical or continuous-time representations of transport maps; here the transport is represented by the flow map of a system of ordinary differential equations (ODEs) or by the corresponding evolution of densities under a system of stochastic differential equations. Some models of this kind explicitly employ triangular structure (Heng et al., 2021), but triangular maps have also proven useful in the analysis of more general neural ODE models (Marzouk et al., 2024).

Within this broader landscape, what is the **role of triangular transport**? We believe that the greatest strength of triangular transport lies in its extraordinary capacity for *parsimony*: Triangular maps may not only naturally exploit conditional independence – a special property among transport-based samplers – but they allow us to fine-tune the level of nonlinearity with which we resolve statistical dependencies. In consequence, these methods allow us to tailor the maps precisely to the system's demands by building some structure of the target distribution into the map itself. In the small sample size regime, we believe this property holds the key to challenging the prevalence of linear methods, which are still the state of the art for practical settings from meteorological data assimilation to subsurface parameter inference.

### 5.3  Where to go from here

As a relatively new method, much of the potential of triangular maps has yet to be realized. This includes a number of promising research directions in applications, methodology, and theory. To unlock the full potential of their parsimony, a key effort for the operationalization of triangular transport is the development of efficient **adaptation strategies**. One of the greatest challenges of the parameterizations discussed in this paper is their scaling with the dimension of the target distribution. One way to mitigate this "curse of dimensionality" is to only include terms one knows will help, which is the problem that the adaptive algorithms above attempt to tackle. As briefly discussed in Section 4.2, these algorithms automatically seek to identify parsimonious maps for general inference problems. Increasing the efficiency of these methods will ease their applicability

to higher-dimensional systems. Promising research avenues include exploring connections to graph structure learning algorithms that estimate conditional independence structure (Baptista et al., 2024d; Liaw et al., 2025; Drton & Maathuis, 2017; Zheng et al., 2020) or developing information criteria (Lunde, 2025; Konishi & Kitagawa, 1996) to help identify suitable levels of map complexity.

A closely-related subject is research into the properties, advantages, and drawbacks of different **parameterizations** of triangular transport. While we have explored many practical details in this tutorial, we have still only scratched the surface of this topic. A promising avenue is to focus on known computational bottlenecks of existing parameterizations. As a simple example, it is known the erf function (see Equation (30)) has unstable evaluations near the extremes of its domain and is often computationally expensive; exploring more efficient alternatives is of cross-cutting utility. Similarly, some parameterization heuristics such as centering RBFs at marginal quantiles of the target distribution remain largely ad hoc, especially given strong dependencies in $\pi$; what can we do to improve these heuristics? What kinds of parameterizations should be employed when the distributions at hand are heavy-tailed: do the "edge terms" of maps need to increase sub- or super-linearly (depending on the direction of the map), and does the use of composite maps relieve the need to model tails when performing conditional sampling? More broadly, is there a practical role for *nonparametric* representations of transport maps (Pooladian & Niles-Weed, 2021) in high-dimensional problems? And whether in parametric or nonparametric settings, how can *regularization* or penalization schemes for learning maps from limited information be more rigorously designed and scaled? Relatedly, it is common nowadays to have multi-fidelity data sources (e.g., models that can be resolved at different discretization levels); how can we efficiently learn maps via samples and model evaluations of differing fidelities?

Furthermore, there are many open questions at the intersection of triangular transport with numerical analysis, relevant to probabilistic modelling and uncertainty quantification. For example, there has been recent work marrying deterministic quadrature or quasi-Monte Carlo methods with transport (Cui et al., 2025a; Klebanov & Sullivan, 2023; Liu, 2024), promising higher-order convergence rates for the approximation of expectations weighted by complex distributions. It is interesting to consider how triangular structure could be leveraged in this setting and to understand broader links to cubature (Cools, 1997) or other high-dimensional integration schemes.

Finally, as we have argued that sparsity provides a route to scalability, it is important to understand the impact of *approximate* sparsity in transport, and hence *approximate* conditional independence in continuous non-Gaussian distributions – e.g., the error incurred by discarding weak conditional dependencies. (See Johnson & Willsky (2007); Jog & Loh (2015) for analyses in the Gaussian case.) This line of work has direct links to *localization* methods widely used in data assimilation, whose analysis is a topic of much current research (Al-Ghattas & Sanz-Alonso, 2024; Tong & Morzfeld, 2023; Gottwald & Reich, 2024). Moreover, sparsity is only one form of low dimensionality. Other work seeks scalable inference and probabilistic modelling via explicit dimension reduction – for instance, by searching for low-dimensional subspaces that capture important interactions between parameters and data, or by identifying low-dimensional "updates" from prior to posterior in the Bayesian setting (Baptista et al., 2022; Zahm et al., 2022; Constantine, 2015; Fukumizu et al., 2007; Härdle & Stoker, 1989; Cui et al., 2014). These notions correspond naturally to specific transport representations (Brennan et al., 2020; Cao et al., 2024; Cui et al., 2025b) and enable the solution of problems that would otherwise – without the exploitation of structure – be intractable.

### 5.4 Toolboxes

We hope that this tutorial provided an accessible introduction to the theory and implementation of triangular transport. As we conclude, we want to leave you, the reader, with some concrete numerical tools to explore your own applications of triangular transport – if you do not want to code your own. The code to reproduce the figures and examples in this tutorial is available on (redacted for anonymization of the review). Further toolboxes, normally listed below, are similarly redacted for the review process.

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

## A  Appendix 1: Optimizing linear separable maps

As established in Section 3.1.2, choosing a *linear separable* map parameterization permits more efficient map optimization. First, for the purpose of this material we denote gradient of some scalar-valued function $r(\mathbf{z})$ with respect to $\mathbf{z}$ evaluated at $\mathbf{z}^*$ as $\nabla_{\mathbf{z}} r\big|_{\mathbf{z}^*}$ and the derivative of a univariate function $s(z)$ with respect to scalar $z$ evaluated at $z^*$ as $\partial_z s\big|_{z^*}$. Then, recall for a separable map that is *linear in the coefficients*, each map component function $S_k$ is defined as:

$$
\begin{aligned}
S_k(x_1, \ldots, x_{k-1}, x_k) &= g(\mathbf{x}_{1:k-1}) + f(x_k) \\
&= c_{k,1}^{\mathrm{non}} \psi_{k,1}^{\mathrm{non}}(\mathbf{x}_{1:k-1}) + \cdots + c_{k,m}^{\mathrm{non}} \psi_{k,m}^{\mathrm{non}}(\mathbf{x}_{1:k-1}) + c_{k,1}^{\mathrm{mon}} \psi_{k,1}^{\mathrm{mon}}(x_k) + \cdots + c_{k,n}^{\mathrm{mon}} \psi_{k,n}^{\mathrm{mon}}(x_k) \\
&= \boldsymbol{\Psi}_k^{\mathrm{non}}(\mathbf{x}_{1:k-1}) \mathbf{c}_k^{\mathrm{non}} + \boldsymbol{\Psi}_k^{\mathrm{mon}}(x_k) \mathbf{c}_k^{\mathrm{mon}}
\end{aligned}
\tag{51}
$$

where $\psi_{k,j}^{\mathrm{non}} : \mathbb{R}^{k-1} \to \mathbb{R}$ and $\psi_{k,j}^{\mathrm{mon}} : \mathbb{R} \to \mathbb{R}$ are the $j$-th basis functions of map component $S_k$, associated with the nonmonotone and monotone terms, respectively. Then, $\boldsymbol{\Psi}_k^{\mathrm{non}} : \mathbb{R}^{k-1} \to \mathbb{R}^{1 \times m}$ and $\boldsymbol{\Psi}_k^{\mathrm{mon}} : \mathbb{R} \to \mathbb{R}^{1 \times n}$ are vectors of basis function evaluations for $g$ and $f$, and $\mathbf{c}_k^{\mathrm{non}} \in \mathbb{R}^{m \times 1}$ and $\mathbf{c}_k^{\mathrm{mon}} \in \mathbb{R}^{n \times 1}$ are the corresponding column vectors of coefficients. Therefore, the expressions $\boldsymbol{\Psi}_k^{\mathrm{non}}(\mathbf{x}_{1:k-1}) \mathbf{c}_k^{\mathrm{non}}$ and $\boldsymbol{\Psi}_k^{\mathrm{mon}}(x_k) \mathbf{c}_k^{\mathrm{mon}}$ are both inner product functions of $\mathbf{x}$, i.e. scalar-valued. For *maps from samples* (Section 3.2.1), we consider samples $\mathbf{X}^1, \ldots, \mathbf{X}^N \sim \pi$ and recall the following optimization objective for $S_k$:

$$
\mathcal{J}_k(S_k) = \sum_{i=1}^N \left( \frac{1}{2} S_k(\mathbf{X}^i)^2 - \log \partial_{x_k} S_k\big|_{\mathbf{X}^i} \right)
\tag{52}
$$

Plugging in Equation 51, we obtain

$$
\begin{aligned}
\mathcal{J}_k(S_k) &= \sum_{i=1}^N \frac{1}{2} \left[ \boldsymbol{\Psi}_k^{\mathrm{non}}(\mathbf{X}_{1:k-1}^i) \mathbf{c}_k^{\mathrm{non}} + \boldsymbol{\Psi}_k^{\mathrm{mon}}(\mathbf{X}_k^i) \mathbf{c}_k^{\mathrm{mon}} \right]^2 - \log \partial_{x_k} \left[ \boldsymbol{\Psi}_k^{\mathrm{mon}}(\mathbf{X}_{1:k-1}^i) \mathbf{c}_k^{\mathrm{non}} + \boldsymbol{\Psi}_k^{\mathrm{mon}}(\mathbf{X}_k) \mathbf{c}_k^{\mathrm{mon}} \right]_{\mathbf{X}_k^i} \\
&= \sum_{i=1}^N \frac{1}{2} \left[ \boldsymbol{\Psi}_k^{\mathrm{non}}(\mathbf{X}_{1:k-1}^i) \mathbf{c}_k^{\mathrm{non}} + \boldsymbol{\Psi}_k^{\mathrm{mon}}(\mathbf{X}_k^i) \mathbf{c}_k^{\mathrm{mon}} \right]^2 - \log \partial_{x_k} \boldsymbol{\Psi}_k^{\mathrm{mon}}\big|_{\mathbf{X}_k^i} \mathbf{c}_k^{\mathrm{mon}}.
\end{aligned}
\tag{53}
$$

Note that the samples $\mathbf{X}^i$ are defined by the target distribution $\pi$, and are thus fixed during optimization. In consequence, the basis function evaluation vectors $\boldsymbol{\Psi}_k^{\mathrm{non}}$ and $\boldsymbol{\Psi}_k^{\mathrm{mon}}$ are also fixed for a given map parameterization (see Section 3.1.2), and are thus independent of the coefficients $\mathbf{c}_k^{\mathrm{non}}$ and $\mathbf{c}_k^{\mathrm{mon}}$. We can simplify Equation 53 further by absorbing the sum over the first term, and defining a new variable for the partial derivative in the second term. To this end, we form matrices $\mathbf{P}_k^{\mathrm{non}} \in \mathbb{R}^{N \times m}$ and $\mathbf{P}_k^{\mathrm{mon}} \in \mathbb{R}^{N \times n}$, and vectors $\mathbf{b}_k^i \in \mathbb{R}^{1 \times n}$, which are each defined element-wise as

$$
[\mathbf{P}_k^{\mathrm{non}}]_{ij} = \psi_{k,j}^{\mathrm{non}}(\mathbf{X}_{1:k-1}^i), \quad [\mathbf{P}_k^{\mathrm{mon}}]_{ij} = \psi_{k,j}^{\mathrm{mon}}(\mathbf{X}_k^i), \quad [\mathbf{b}_k^i]_j = \partial_{x_k} \psi_{k,j}^{\mathrm{mon}}\big|_{\mathbf{X}_k^i},
$$

where the $ij$ entry is the $j$th indexed basis function evaluated at sample index $i$ for both matrices $\mathbf{P}_k^{\mathrm{non}}$ and $\mathbf{P}_k^{\mathrm{mon}}$, and the log of the derivative of $\boldsymbol{\Psi}_k^{\mathrm{mon}}$ with respect to $\mathbf{X}_k$ is evaluated at $\mathbf{X}_k^i$ for the vectors $\mathbf{b}_k^i$. As above, these matrices and vectors can be pre-computed. The optimization objective now simplifies further to

$$
\mathcal{J}_k(S_k) = \frac{1}{2} \left\| \mathbf{P}_k^{\mathrm{non}} \mathbf{c}_k^{\mathrm{non}} + \mathbf{P}_k^{\mathrm{mon}} \mathbf{c}_k^{\mathrm{mon}} \right\|^2 - \sum_{i=1}^N \log \mathbf{b}_k^i \mathbf{c}_k^{\mathrm{mon}}.
$$

This function is similar to that of an objective for an interior point method and, remarkably, becomes quadratic in $\mathbf{c}_k^{\mathrm{non}}$. At this point, we add L2 (i.e., Tikhonov) regularization on both $\mathbf{c}_k^{\mathrm{mon}}$ and $\mathbf{c}_k^{\mathrm{non}}$ according to the guidance in Section 3.

$$
\mathcal{J}_k(S_k; \lambda) = \frac{1}{2} \left\| \mathbf{P}_k^{\mathrm{non}} \mathbf{c}_k^{\mathrm{non}} + \mathbf{P}_k^{\mathrm{mon}} \mathbf{c}_k^{\mathrm{mon}} \right\|^2 - \sum_{i=1}^N \log \mathbf{b}_k^i \mathbf{c}_k^{\mathrm{mon}} + \frac{\lambda}{2} (\|\mathbf{c}_k^{\mathrm{non}}\|^2 + \|\mathbf{c}_k^{\mathrm{mon}}\|^2)
\tag{54}
$$

In practice, this means the optimal coefficients $\widehat{\mathbf{c}}_k^{\text{non}}$ for the nonmonotone basis function evaluations minimizing (54) must satisfy

$$
\begin{aligned}
0 &\equiv \nabla_{\mathbf{c}_k^{\text{non}}} \mathcal{J}_k(S_k; \lambda)\big|_{\widehat{\mathbf{c}}_k^{\text{non}}} \\
&= \nabla_{\mathbf{c}_k^{\text{non}}} \frac{1}{2} \| \mathbf{P}_k^{\text{non}} \mathbf{c}_k^{\text{non}} + \mathbf{P}_k^{\text{mon}} \mathbf{c}_k^{\text{mon}} \|^2 \big|_{\widehat{\mathbf{c}}_k^{\text{non}}} + \lambda \mathbf{c}_k^{\text{non}} \\
&= (\mathbf{P}_k^{\text{non}\top} \mathbf{P}_k^{\text{non}} + \lambda \mathbf{I}) \widehat{\mathbf{c}}_k^{\text{non}} + \mathbf{P}_k^{\text{non}\top} \mathbf{P}_k^{\text{mon}} \mathbf{c}_k^{\text{mon}}.
\end{aligned}
\tag{55}
$$

In consequence, for a given choice of coefficients parameterizing the monotone functions, $\mathbf{c}_k^{\text{mon}}$, we can find the optimal choice of $\widehat{\mathbf{c}}_k^{\text{non}}$ by solving the *normal equations*; for this scenario, we assume that $m < N$, i.e. the number of samples surpasses the number of basis functions (and thus $\mathbf{P}_k^{\text{non}\top} \mathbf{P}_k^{\text{non}}$ is full rank). The normal equations are a well-studied class of problems in numerical linear algebra (Trefethen & Bau, 2022). Assuming linearly independent basis functions, this affords a solution for $\widehat{\mathbf{c}}_k^{\text{non}}$ as a function of $\mathbf{c}_k^{\text{mon}}$ given as

$$
\begin{aligned}
(\mathbf{P}_k^{\text{non}\top} \mathbf{P}_k^{\text{non}} + \lambda \mathbf{I}) \widehat{\mathbf{c}}_k^{\text{non}} &= -\mathbf{P}_k^{\text{non}\top} \mathbf{P}_k^{\text{mon}} \mathbf{c}_k^{\text{mon}} \\
\widehat{\mathbf{c}}_k^{\text{non}} &= -\underbrace{(\mathbf{P}_k^{\text{non}\top} \mathbf{P}_k^{\text{non}} + \lambda \mathbf{I})^{-1} \mathbf{P}_k^{\text{non}\top}}_{\mathbf{M}_{k,\lambda}} \mathbf{P}_k^{\text{mon}} \mathbf{c}_k^{\text{mon}} \\
&\overset{\text{def}}{=} -\mathbf{M}_{k,\lambda} \mathbf{P}_k^{\text{mon}} \mathbf{c}_k^{\text{mon}}
\end{aligned}
\tag{56}
$$

Following best practices from numerical linear algebra, this inversion should not be computed explicitly; rather, one should use a numerical solver for the systems induced. Then, substituting Expression (56) into the optimization objective, we obtain a new objective for $\mathbf{c}_k^{\text{mon}}$ (i.e., entirely independent from the nonmonotone coefficients),

$$
\begin{aligned}
\mathcal{J}_k^{\text{mon}}(\mathbf{c}_k^{\text{mon}}; \lambda) &= \frac{1}{2} \| -\mathbf{P}_k^{\text{non}} \mathbf{M}_{k,\lambda} \mathbf{P}_k^{\text{mon}} \mathbf{c}_k^{\text{mon}} + \mathbf{P}_k^{\text{mon}} \mathbf{c}_k^{\text{mon}} \|^2 - \sum_{i=1}^N \log \mathbf{b}_k^i \mathbf{c}_k^{\text{mon}} + \frac{\lambda}{2} (\|\widehat{\mathbf{c}}_k^{\text{non}}\|^2 + \|\mathbf{c}_k^{\text{mon}}\|^2) \\
&= \frac{1}{2} \| \underbrace{(\mathbf{I} - \mathbf{P}_k^{\text{non}} \mathbf{M}_{k,\lambda}) \mathbf{P}_k^{\text{mon}}}_{\mathbf{A}_{k,\lambda}} \mathbf{c}_k^{\text{mon}} \|^2 - \sum_{i=1}^N \log \mathbf{b}_k^i \mathbf{c}_k^{\text{mon}} + \frac{\lambda}{2} (\| \underbrace{\mathbf{M}_{k,\lambda} \mathbf{P}_k^{\text{mon}}}_{\mathbf{D}_{k,\lambda}} \mathbf{c}_k^{\text{mon}} \|^2 + \|\mathbf{c}_k^{\text{mon}}\|^2) \\
&\overset{\text{def}}{=} \frac{1}{2} \| \mathbf{A}_{k,\lambda} \mathbf{c}_k^{\text{mon}} \|^2 - \sum_{i=1}^N \log \mathbf{b}_k^i \mathbf{c}_k^{\text{mon}} + \frac{\lambda}{2} (\| \mathbf{D}_{k,\lambda} \mathbf{c}_k^{\text{mon}} \|^2 + \|\mathbf{c}_k^{\text{mon}}\|^2),
\end{aligned}
\tag{57}
$$

where $\mathbf{A}_{k,\lambda}$, $\mathbf{D}_{k,\lambda}$ and $\mathbf{b}_k^i$ can be precomputed prior to the optimization routine via evaluation of the basis functions. Remarkably, this method translates the original loss function into a very simple constrained convex optimization problem

$$
\boxed{\widehat{\mathbf{c}}_k^{\text{mon}} = \underset{\mathbf{c}_k^{\text{mon}} \geq \mathbf{0}}{\arg\min} \frac{1}{2} \| \mathbf{A}_{k,\lambda} \mathbf{c}_k^{\text{mon}} \|^2 - \sum_{i=1}^N \log \mathbf{b}_k^i \mathbf{c}_k^{\text{mon}} + \frac{\lambda}{2} (\| \mathbf{D}_{k,\lambda} \mathbf{c}_k^{\text{mon}} \|^2 + \|\mathbf{c}_k^{\text{mon}}\|^2), \quad \widehat{\mathbf{c}}_k^{\text{non}} = -\mathbf{M}_{k,\lambda} \mathbf{P}_k^{\text{mon}} \widehat{\mathbf{c}}_k^{\text{mon}}.}
$$

$$\tag{58}$$

Note that the (element-wise) constraint of $\mathbf{c}_k^{\text{mon}} \geq \mathbf{0}$ is vital to maintain monotonicity; this can be enforced explicitly during optimization using particular optimization algorithms, e.g., L-BFGS-B (i.e. Low-memory BFGS with box constraints), implicitly by constructing an objective with a log-barrier term, or employing a convex transformation of the optimization objective (e.g. optimize over $\mathbf{p}_k^{\text{mon}} := \log \mathbf{c}_k^{\text{mon}}$). Since we often will have coefficients with zero values, the first methodology might be preferable (though no empirical results appear here). This optimization objective, notably, requires no evaluations of the map during optimization, that is to say, the optimization is as fast as the combination of the implementation of linear algebra algorithms called and the optimization routine used.

In the case where we have more parameters than samples, it is worth noting that using this formulation is remarkably sensitive to $\lambda$ because $\mathbf{P}_k^{\text{non}\top} \mathbf{P}_k^{\text{non}}$ is no longer full rank. Thus the calculation of $\mathbf{M}_{k,\lambda}$ solves

a system with possibly poor numerical properties (the industry of methods for *ill-conditioned* systems is dedicated to such problems). For vanishingly small $\lambda$, this corresponds to the problem of overfitting and the fact that we have infinite choices of $\widehat{\mathbf{c}}_k^{\mathrm{non}}$ for any given choice of $\mathbf{c}_k^{\mathrm{mon}}$.

