# OpenReview forum: "A friendly introduction to triangular transport"
_TMLR — Under review for TMLR_

### Review · Reviewer_dGAx · 2026-06-08

**Summary Of Contributions:**

This article provides an introduction to triangular transport. After introducing the problem and expliciting some applications of these models, the authors clearly explain and describe how triangular transport models work theoretically. Then, they explain in greater detail how to construct and optimize the transport function in practice. Finally, they discuss the potential limitations and challenges of these models.

This article is very well written and is easily accessible to anyone without a strong background in theoretical mathematics in my opinion. It clearly lives up to what is promised at the start.

**Additional Comments:**

None

**Audience:**

Yes

**Audience Explanation:**

This article is strongly in the scope of the TMLR journal.

**Claims And Evidence:**

Yes

**Claims Explanation:**

Everything is clearly explained.

**Requested Changes:**

I have some comments and remarks about this article.

First, in Section 2.3, the authors claim that "Over all functions satisfying this structure, there is one that uniquely couples π and η under mild conditions". I would have appreciated more details about it. In particular, is the existence of the transformation always guaranteed? And is the unicity valid only for a given permutation?

Second, since Equation (8) is one of the most important equation of the paper, I would add at least some hints of the proof, as well as some additional explanations on why it is valid if and only if the reference distribution has independant marginals. Moreover, does the ordering of the variables impact the conditional generation? In other words, is it possible to sample from $x_1|x_2$?

Third, in Section 2.3.3, the authors highlight that conditional independance can strongly decrease the complexity of the transport. Then, I was wondering if the conditionnal independance between some variables needs to be known a priori, or can it be identified during training?

At the beginning of Section 3, some hypotheses need to be more explained. Indeed, why is it important that $\eta$ and $\pi$ share the same mean and covariance? If it is not the case, we do it empirically with a sample from $\pi$? In addition, why do we need to assume that $\eta$ and $\pi$ have approximately the same tails?

In Section 3.1.1, I was wondering why the authors only consider polynomial functions for $S^{non}$. Is it only for illustrative purposes, or because these functions are expressive enough? As a consequence, I don't really understand the link with Section 3.1.2.
To sum up, I would suggest to explain more clearly the structure of Section 3.1 at the beginning such that the reader can follow the ideas more easily.

I have also some more global remarks:
- The authors sometimes mention the problem of the dimension. I think it would have been interesting to have an order of magnitude of the maximum acceptable dimension when using the full transformation, and same when using with sparser transformations.
- In my opinion, this article lacks of a real test case section at the end to compare numerically the performances of triangular transport with NFs and/or GANs for example (test case of cats as in the beginning, or MNIST).
- In Section 4, I would have added pseudo-codes to illustrate the practical implementations of the proposed methods.

Minor remarks:
- Ensure consistency in punctuation, in particular before equations (I suggest to add ":" before every equation), and points at the end of equations if required (equations (4)(13)(25) for example).
- Add a space after "Equation (1)" in Section 2.1
- In Section 2.2, maybe add that $\eta$ and $\pi$ must have the same dimension (I think it is not clearly said in the article)
- At the very beginning of  Section 2.3, the authors say that GAN can be a transport between $\eta$ and $\pi$. However, to the best of my understanding, GANs are a little bit different that NFs and triangular transports since they don't need that both distributions share the same dimension. Then, I think it is a little bit misleading to compare NFs and triangular transports with GANs.
- There is a typo in Equation (7): it is $z_3$ instead of $z_2$ in the third line on the transformation.

---

### Review · Reviewer_Z7FJ · 2026-07-14

**Summary Of Contributions:**

This submission provides a detailed and accessible introduction to the triangular transport framework which enables modelling a target distribution $\pi$ in terms of a transformation of a simple reference distribution $\eta$. Interestingly, triangular transport can also be used to characterize the conditional distribution of $\pi$ so that this method can be applied for Bayesian inference. The work begins by describing the main attributes of  triangular transport, namely parsimony (the ability to adjust the complexity of the map), sparsity (the ability to exploit conditional independence), numerical convenience (the method is easily implementable), and explainability (the maps give a clear correspondence between their components and the corresponding statistical features). Given these useful features, triangular transport has been applied to Bayesian inference, data assimilation, density estimation and generative modelling, and optimal experimental design among other applications.

With this, the work proceeds with a description of the theoretical underpinnings of the method. Effectively, the idea is to leverage the pushforward/pullback formulas and to find the map which best connects $\pi$ and $\eta$ among the class of "triangular maps" which consist of maps from $\mathbb R^k\to\mathbb R$ with $k$ map components $S_1,\dots,S_k$ where $S_1$ acts only on the first coordinate, $S_2$ acts only on the first two coordinates, and so on. The map should also be monotone in a certain sense. Such a map exists under mild conditions, is easily invertible, and provide a factorization of $\pi$ in terms of the marginal conditional pdf, enabling the practitioner to sample from the conditionals.

Given these theoretical justifications, the remainder of the paper concerns implementation of the method and practical concerns. First, it is described how the required monotonicity can be ensured using a number of different approaches, each with their own strengths and limitations. Next, the practical optimization problem to be solved is provided in two settings (1) only samples are provided from the target pdf (2) the target pdf is known, but is not normalized. A short discussion of regularization is also provided.

The submission concludes with some discussion of potential research directions on triangular transport including improving the efficiency of these methods in high-dimensional regimes, improving our understanding of the different parametrizations, and analyzing the effects of sparsity.

**Additional Comments:**

I do not believe that the current submission falls within the scope of TMLR.

The paper is not exactly a survey, since it focuses only on non-technical aspects of the area and does not clearly indicate where the different ideas discussed stem from. In my opinion, this submission is closer to a chapter in an explanatory book which does not appear to fall within the submission guidelines.

Furthermore, some of the figures in the work are copied with only minor modifications from other papers without attribution. For instance, Figure 5 is a minor edit of Figure 3 from [1] and, similarly, Figure 6 is a minor edit of Figure 1 from [2]. Even if these papers were originally written by the authors, the editorial policies explicitly prohibit the reuse of written text, figures, or results between the submitted paper and any paper which has been published.

[1] Ramgraber, M., Baptista, R., McLaughlin, D., & Marzouk, Y. (2023). Ensemble transport smoothing. Part I: Unified framework. Journal of Computational Physics: X, 17, 100134.

[2] Ramgraber, M., Baptista, R., McLaughlin, D. and Marzouk, Y. (2023). Ensemble transport smoothing. Part II: Nonlinear updates. Journal of Computational Physics: X, 17, p.100133.

**Audience:**

Yes

**Audience Explanation:**

I believe the manuscript is suitable for those who would be interested in learning more about triangular transport, but only if they are looking for a general view on the method without much theoretical overhead.

**Broader Impact Concerns:**

I have no concerns with the ethical implications of the work.

**Claims And Evidence:**

Yes

**Claims Explanation:**

To my understanding, the submission does not provide any new claims on top of what is in the existing literature. Most of the points made are justified via a reference or some explanation.

**Requested Changes:**

1. I find the use of color in the main text to be nonstandard and it does not add much to the paper in my opinion. I would recommend to keep the main text color black.

2. A reference for the pullback/pushforward formulas should be included.

3. Point 1 on p.8 should be expanded. What are the mild conditions guaranteeing the existence of a triangular map? Does a triangular map satisfying the required monotonicity property necessarily exist?

---

### Review · Reviewer_SVsb · 2026-07-17

**Summary Of Contributions:**

This tutorial introduces triangular transport maps. It is addressed to applied researchers who wish to use measure
transport for generative modelling and Bayesian inference without a formal and heavy mathematical background. It develops the theory in intuition-first fashion (Bayes' theorem, the change-of-variables formula, the triangular structure and its factorization of the joint density, conditional sampling, and the link between conditional independence and map sparsity). Then, it focuses on implementation aspects in depth including modeling and optimization/learning. It closes with practical heuristics, a broad literature outlook, and an appendix deriving a closed-form solution for the nonmonotone coefficients of a linear-separable map. The paper's main contribution is
pedagogical: a careful, unusually well-illustrated synthesis of scattered implementation knowledge, rather than a new method or result.

**Audience:**

Yes

**Audience Explanation:**

Definitely yes. Triangular transport underpins methods in data assimilation,
inverse problems, and simulation-based inference. The practitioners in those
communities can surely benefit from this accessible and implementation-focused treatment.

**Broader Impact Concerns:**

No broader impact concerns.

**Claims And Evidence:**

Yes

**Claims Explanation:**

Broadly yes. As a tutorial, the claims are largely expository, and the mathematics is sound and clearly illustrated. My reservations concern presentation and the way some specific statements have been made. Some examples:

(1) pp. 9-11: the conditional-sampling and sparsification machinery (Eqs.~10--16) quietly assumes the conditioning set is a
    prefix $x_{1:k}$ of the chosen ordering. A fixed map provides only
    the conditionals of a suffix given a prefix; other conditionals require a
    different ordering (hence a different map). This important limitation, and
    the fact that sparsity depends on the ordering, are stated confidently in
    the early bullets but only qualified later.

(2) pp. 12: ``sparse maps can overcome this dramatic increase in
    complexity'' is true only when the conditional independence holds (at least
    approximately); otherwise sparsification injects bias. The outlook (5.3)
    acknowledges approximate CI, but the confident p.~12 claim is not linked to
    it.

**Requested Changes:**

(Critical) A few related places where the exposition could be made clearer or more balanced: (i) On p. 5, moving between the joint distribution and its conditionals (and back) comes across as convoluted for this class of models. A little more explanation of why this is the case would help the reader appreciate the strengths and weaknesses of the approach. (ii) It would help to state clearly that a fixed map yields only suffix-given-prefix conditionals, and that sparsity is ordering-dependent. Forward-referencing the p. 11 caveats at first mention (pp. 2, 9–11) would be helpful. (iii) The p. 12 sparsity claim could be gently hedged to note the bias incurred when conditional independence holds only approximately. (iv, minor) It would be nice to reserve "approximates" for the learned-map setting (p. 6).

(Critical) The case of partially observed data is of interest for many practitioners. The maps-from-samples framework appears to assume complete samples Xi∼πX^i \sim \pi Xi∼π. How does one train and perform inference when samples have missing entries, particularly under non-systematic missingness (different entries absent across samples)? Does the triangular structure help or hinder, and do standardization (Section 3.1.3) and the closed-form optimization (Appendix A) still apply? Even a brief discussion would help, as missing data is common in the advertised application domains.

(Critical) It would help to show how the learned map supports downstream operations beyond sampling, such as marginalization, computing expectations/integrals, entropy, and mutual information. A short recipe for obtaining these from the fitted map would add real practical value for readers.

(Suggested) It might be nice to add some explicit scalability and complexity ranges. The paper argues sparsity is the key to high-dimensional applicability, but the cost discussion is qualitative ("scales linearly in K," "the demand explodes"). Providing big-O complexity for the core operations, like forward evaluation, sequential inversion, and per-component optimization, as functions of the target dimension K, sample size N, and number of basis functions/parameters would be a helpful addition for the readers.

(Suggested) Defining "triangular" where it is first used (p. 2), ideally by previewing the joint-density factorization π(x)=π(x1)π(x2∣x1)⋯π(xK∣x1:K−1) that currently appears only at Eq. (8), would be helpful in my opinion. In the same spirit, defining "multivariate monotone function" at first use on p. 6, and giving a minimal graph example alongside the "Sparsity" bullet on p. 2 to make concrete between which variables conditional independence is being exploited, would also help.

(Suggested) A few local expositions could be polished:
The opening of Section 4.1 would read more smoothly with a one-paragraph problem statement before Eq. (50): an imperfect map means naive conditional inversion samples the wrong distribution, which composite maps then fix. It would also help to distinguish x^*_{1:k} from x_{1:k}​. The remark that "we may not be certain about the quality of our map approximation" (p. 31) could use a few concrete diagnostics a practitioner might actually use. The aside on p. 16 about making Eq. (22) separable in xk​ left me wondering why this is true. It would be nice to explain why you chose to cover polynomials and RBFs in your tutorial. Are these the two most popular approaches? Connecting marginal (diagonal) maps to copula-style marginal transformations (p. 15) might resonate with readers from statistics.

(Suggested) Adding experimental results on a couple of benchmark datasets, compared against standard methods (e.g. normalizing flows, optimal-transport-based samplers, VAEs, or MCMC where appropriate) would round out the presentation of the method. The paper's comparative and scalability claims currently rest on low-dimensional illustrative examples. Even a small set of results on realistic or standard problems, with baselines, would substantiate these claims and strengthen the contribution.

(Suggested) Minor line items:
Fix "computationally expensive expensive" (p. 6). Define GANs (and check first-use expansion of other acronyms). Equations (3)–(4) are exact change-of-variables identities, yet the pushforward is said to "approximate" η\eta η. "Generally impossible to formulate p(a,b)p(a,b) p(a,b) in closed form" is confusing, given that the context is learning models.